# Reversible bipolar thermopower of ionic thermoelectric polymer composite for cyclic energy generation

Cheng Chi ®[1,2,5], Gongze Liu[3,5], Meng An ®[1,4,5], Yufeng Zhang[1], Dongxing Song[1], Xin Qi[1], Chunyu Zhao[1], Zequn Wang[4], Yanzheng Du[1], Zizhen Lin[1], Yang Lu[1], He Huang[3], Yang Li[3], Chongjia Lin ®[3], Weigang Ma ®[1]✉, Baoling Huang[3]✉, Xiaoze Du[2] & Xing Zhang[1]

The giant thermopower of ionic thermoelectric materials has attracted great attention for waste-heat recovery technologies. However, generating cyclic power by ionic thermoelectric modules remains challenging, since the ions cannot travel across the electrode interface. Here, we reported a reversible bipolar thermopower (+20.2 mV K⁻¹ to −10.2 mV K⁻¹) of the same composite by manipulating the interactions of ions and electrodes. Meanwhile, a promising ionic thermoelectric generator was proposed to achieve cyclic power generation under a constant heat course only by switching the external electrodes that can effectively realize the alternating dominated thermodiffusion of cations and anions. It eliminates the necessity to change the thermal contact between material and heat, nor does it require re-establish the temperature differences, which can favor improving the efficiency of the ionic thermoelectrics. Furthermore, the developed micro-thermal sensors demonstrated high sensitivity and responsivity in light detecting, presenting innovative impacts on exploring next-generation ionic thermoelectric devices.

Ionic-conducting polymer composite ionic thermoelectric ($i$-TE) materials have attracted wide attention as they exhibited high ionic thermopower or Seebeck coefficient[1–3]. Different from electrons and holes in the electrical-conducting thermoelectric ($e$-TE) materials, the charge carriers of the $i$-TE materials are free cations and anions. Under a temperature gradient, ions will move across the $i$-TE material from the hot side to the cold side. Due to the mass transportation difference between cations and anions, they build an internal ionic Seebeck voltage based on the Soret effect[4,5]. Moreover, the introduced more interactions between ions and polymers significantly enhanced the change in Eastman entropy of polymer composites[6–8], favoring

obtaining a high ionic thermopower or Seebeck coefficient ($S_i$) of ~10 mV K⁻¹, which is about two orders higher than that of semiconductor $e$-TE materials (~200 µV K⁻¹)[9–11]. Moreover, polymer composites essentially exhibit the advantages of flexibility, low cost, and environment friendly, which are regarded as the most promising candidates for low-grade heat harvesting, thermal sensing, and sustainable energy developments[12–15].

Usually, it is necessary to connect p- and n-type $e$-TE materials in series to build TE modules to generate continuous electricity and maximum output power in practical applications. Regarding the emerging $i$-TE materials based on the Soret

¹Key Laboratory for Thermal Science and Power Engineering of Ministry of Education, Department of Engineering Mechanics, Tsinghua University, 100084 Beijing, China. ²Key Laboratory of Power Station Energy Transfer Conversion and System of Ministry of Education, School of Energy Power and Mechanical Engineering, North China Electric Power University, 102206 Beijing, China. ³Department of Mechanical and Aerospace Engineering, The Hong Kong University of Science and Technology, Clear Water Bay, Hong Kong SAR, China. ⁴College of Mechanical and Electrical Engineering, Shaanxi University of Science and Technology, 710021 Xi'an, China. ⁵These authors contributed equally: Cheng Chi, Gongze Liu, Meng An. ✉e-mail: maweigang@tsinghua.edu.cn; mebhuang@ust.hk

effect[16,17], ions show preferential movement along or against thermal gradients, and the ionic thermoelectric capacitor (*i*-TEC) was developed to convert heat to electricity[5,18,19]. However, the *i*-TEC cells cannot produce a continuous current flow under a constant temperature difference (Δ*T*). Because they operated in a capacitive mode and ions only accumulated at the interface near electrodes, ions cannot transport across the electrode interface[20–22]. The heat source must be repeatedly established and removed for every charging and discharging cycle to ensure ions move back and forth, which is not convenient in practical applications. Although the *i*-TE devices periodically contacted the heat and cold source providing another way to convert heat to power, the ionic thermoelectric generators (*i*-TEGs) still need to detach from the heat sources and take time to re-establish the temperature difference in every cycle, which increases energy consumption and reduce the conversion efficiency. Further efforts are needed to address the limitations of ionic thermoelectric technology applications.

Achieving the ions to move back and forth alternately under a constant heat source is a major milestone to achieve producing cyclic energy by *i*-TEGs. Generally, the thermopower is an intrinsic property of *i*-TE material, and the *i*-TE material usually exhibits only one type of thermopower, either p- or n-type characteristics, which is determined by the charge type of ions that dominates the thermodiffusion process. Some recent work reported that the sign of thermopower could be tuned to the opposite by modifying the internal structure or compositions by doping methods[23–25], however, the process of tuning the sign of thermopower can only be converted once and is not reversible. Unluckily, the ions of *i*-TE materials cannot have reciprocating motion behaviors under a constant heat source, up to now, producing continuous energy remains challenging.

Here, we reported a unique reversible bipolar thermopower characteristic of the same solid-state *i*-TE materials by manipulating external electrodes. Specifically, the same polymer composite exhibited a p-type thermopower of +20.2 ± 4 mV K⁻¹ and an n-type thermopower of −10.2 ± 0.83 mV K⁻¹ when assembling with different electrodes at 298 K and 70% relative humidity (RH). It is found that the strength of the interfacial interaction and the polarization arrangement of ions differed significantly relating to the different electrode materials. Moreover, we proposed a promising *i*-TE generator prototype with the bipolar thermopower property, which successfully achieved producing cyclic energy under a constant heat source only by alternately exchanging electrodes. Meanwhile, a developed micro-thermal sensor exhibited high sensitivity when detecting small fluctuations of heat, bringing significant and innovative impacts on the development of next-generation *i*-TE devices.

## Results
### Bipolar thermopower property
The developed solid-state *i*-TE polymer composite material comprised of poly(vinylidene fluoride-co-hexafluoropropylene) (PVDF-HFP), propylene carbonate (PC), and sodium trifluoromethanesulfonimide (NaTFSI), which was abbreviated as PNP and prepared by the solution casting method (details in Supplementary methods). Here, a series of different electrode materials, aligned carbon nanotube (a-CNT), multi-walled CNT (MWCNT), single-walled CNT (SWCNT), and metal electrodes (Au, Pt, and Cu) are selected to serve as test electrodes. For measuring the thermopower, pieces of rectangular PNP films with identical sizes were cut from one matrix and were suspended on these E|E electrodes, which were named the E|PNP|E systems, *E* = (a-CNT, MWCNT, SWCNT, Au, Pt, and Cu). When starting to apply a series of positive and negative temperature gradients (±Δ*T*), the initial hot and cold terminals of the PNP were connected to the work and counter electrodes of the voltage meter, respectively, which is used to monitor the voltage variations in real-time (Fig. S1, Supplementary

Information). For the Cu|PNP|Cu system (Fig. 1a, top), the sign of the generated voltage between the hot side and cold side (Δ*V*$_{Cu}$ = *V*$_{hot}$−*V*$_{cold}$) is negative in response to a positive Δ*T*, as shown in Fig. 1b (blue curve). It was implied that the Na⁺ cations dominated the thermodiffusion process[2,24,26], performing a p-type characteristic, as illustrated in Fig. 1c (left). Then, a series of temperature differences (Δ*T* = ±6 to ±1 K) were applied to the Cu|PNP|Cu systems. The produced thermoelectric voltage differences of p-type Cu|PNP|Cu systems followed a good linear relationship with the temperature differences, as shown in (Figs. 1d and S2a, Supplementary Information). The thermopower or ionic Seebeck coefficient (*S*$_i$) of *i*-TE material is defined as Eq. (1)[24]

$$S_i = -(V_H - V_C)/(T_H - T_C) \qquad (1)$$

where *V*$_H$ and *V*$_C$ correspond to the voltage of the hot electrode at temperature *T*$_H$ and the cold electrode at temperature *T*$_C$, respectively. By fitting the slope of the measured data, the calculated thermopower of the p-type Cu|PNP|Cu was 20.2 ± 4 mV K⁻¹ at RH 70%, 298 K (Fig. 1e). To test the stability of the Cu electrode, the X-ray photoelectron spectroscopy (XPS) curves of Cu electrodes before and after measurement are overlapped well (Fig. S3, Supplementary Information) and no new peaks were observed on the used Cu electrodes, representing no chemical reaction on the Cu electrode. And there is no color change on the surface of the used Cu electrode and no obvious decay of the produced thermoelectric voltage of the Cu|PNP|Cu system after 30 cycles test, suggesting high stability of the Cu electrodes with the PNP composite. Moreover, the control experiment by using the noble metal Pt and Au electrodes to measure the thermopower of PNP was carried out. The measured thermopower of Pt|PNP|Pt and Au|PNP|Au are 19.3 ± 3.2 and 21.1 ± 3.8 mV K⁻¹, respectively (Figs. 1d, e and S4, Supplementary Information) under the same test condition, which is consistent with the thermopower of Cu|PNP|Cu system.

Interestingly, when the same PNP film was assembled with a-CNT electrodes (Fig. 1a), the sign of the produced Δ*V*$_{a-CNT}$ (Δ*V*$_{a-CNT}$ = *V*$_{hot}$−*V*$_{cold}$) of the a-CNT|PNP|a-CNT system was positive in response to a same positive Δ*T* (red line, Fig. 1b), which is opposite to the sign of the Cu|PNP|Cu system (blue line, Fig. 1b). Accordingly, the positive Δ*V*$_{a-CNT}$ suggested that a higher number of TFSI⁻ anions moved to the cold side[25], and the TFSI⁻ anions dominated the thermodiffusion process of the a-CNT|PNP|a-CNT system (Fig. 1c, right), exhibiting n-type behavior. Thus, the thermal diffusion sequences of Na⁺ and TFSI⁻ ions have been reversed when assembling with a-CNT electrodes compared to Cu electrodes. In addition, after introducing a series of positive and negative temperature gradients (Δ*T* = ± 6 to ±1 K) to the a-CNT|PNP|a-CNT system, the sign of the produced thermal voltage is always opposite to the Cu|PNP|Cu system at every Δ*T* (Fig. S2b, Supplementary Information). After fitting the slope of the measured thermal voltage Δ*V*$_{a-CNT}$ with the Δ*T* (Fig. 1d), the calculated thermopower of n-type a-CNT|PNP|a-CNT systems was −10.2 ± 0.83 mV K⁻¹ (Fig. 1e).

In addition, we also conducted to measure the thermopower of PNP using the SWCNT and MWCNT-based electrodes. They also performed n-type characteristics and the measured thermopower of PNP tested by SWCNTs and MWCNTs are −8.17 ± 1.2 and −5.2 ± 1.35 mV K⁻¹, respectively, (Fig. 1d, e). The digital photos of the a-CNT, MWCNT, and SWCNT films are shown in Fig. S5 (Supplementary Information). Due to their softness and self-adhesive nature, the a-CNT is easy to form tightly contact with the PNP composite compared to thick SWCNT and MWCNT films. Having good contact is of great significance for reducing contact resistance and improving the ionic thermoelectric conversion performance. Meanwhile, the a-CNT bundles are well aligned along the length direction with small entanglements, they exhibited higher conductivity of $3 \times 10^5$ S m⁻¹, a lower sheet resistance of

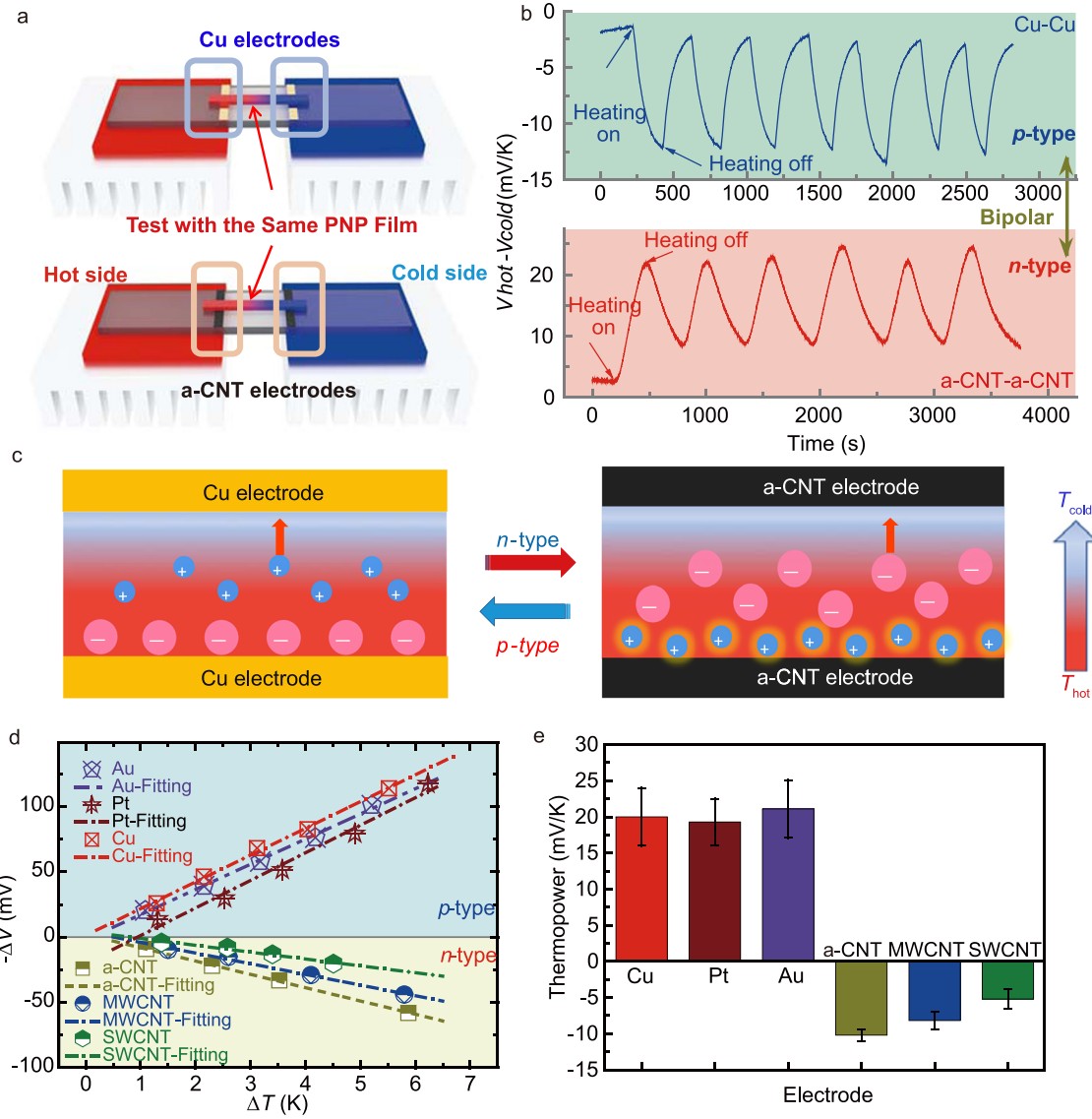

**Fig. 1 | The bipolar thermopower of the same PNP. a** The illustration of the thermopower test using Cu electrodes and aligned CNT (a-CNT) electrodes. **b** The curves of the generated thermoelectric voltage as a function of time for the Cu|PNP|Cu (blue line) and a-CNT|PNP|a-CNT (red line) systems under several alternating heating on/off cycles. **c** The schematic illustration of the thermodiffusion process of the p-type Cu|PNP|Cu and n-type a-CNT|PNP|a-CNT systems under a temperature gradient. **d** The fitting curves of the produced thermoelectric voltage difference corresponding to related temperature differences and **e** the values of the measured thermopowers of the E|PNP|E systems (E = Cu, Pt, Au, a-CNT, MWCNT, and SWCNT), which error bars were calculated using the standard deviation of the measured thermopower data.

1–1.5 Ω m$^{-2}$, a stronger mechanical strength of -120 MPa, low cost and high stability and flexibility compared to common SWCNT and MWCNT-based films, as summarized in Table S1 (Supplementary Information).

Moreover, the thermopowers of the PNP samples at different relative humidity (50% RH–90% RH) were further investigated. The ionic thermopower of the p-type Cu|PNP|Cu system increased from 8.12 mV K$^{-1}$ (50% RH) to 26.62 mV K$^{-1}$ (90% RH), (Fig. S6, Supplementary Information). The porous structure of PVDF-HFP and the hydrophilic nature of sodium salt tend to absorb water from the moisture environment and could fill the space in the PVDF-HFP matrix. The absorbed water could help improve the dissociation of the NaTFSI ion pairs by weakening the electrostatic attraction[6,27]. Further increasing the humidity level may create continuous water percolation paths or ion transport channels, which could reach a saturation state. The magnitude of the ionic thermopower of the n-type a-CNT|PNP|a-CNT sample slightly increased from −8.11 ± 1.5 mV K$^{-1}$ at 50% RH to

−10.2 ± 0.83 at 70% RH, which was caused by water absorption effect as discussed above. Further increasing the humidity level, the thermopower reached a relative saturation value of −9.7 ± 0.86 mV K$^{-1}$ at 80% RH (Fig. S7, Supplementary Information). Thus, varying the humidity level only contributes to the change in the magnitude of the thermopower of Cu|PNP|Cu and a-CNT|PNP|a-CNT systems, but it does not influence the sign of the p-type characteristic of the Cu|PNP|Cu and the n-type behavior of a-CNT|PNP|a-CNT.

**Bipolar thermopower property analysis**
Interestingly, the same PNP material with different electrodes performed p- and n-type characteristics by assembling with different external electrode materials. The above findings strongly suggested the electrode materials induce significant disparities in the transportation kinetics of ions near the electrodes. The p-type Cu|PNP|Cu and n-type a-CNT|PNP|a-CNT were selected to further investigate the electrode effect on the thermopower.

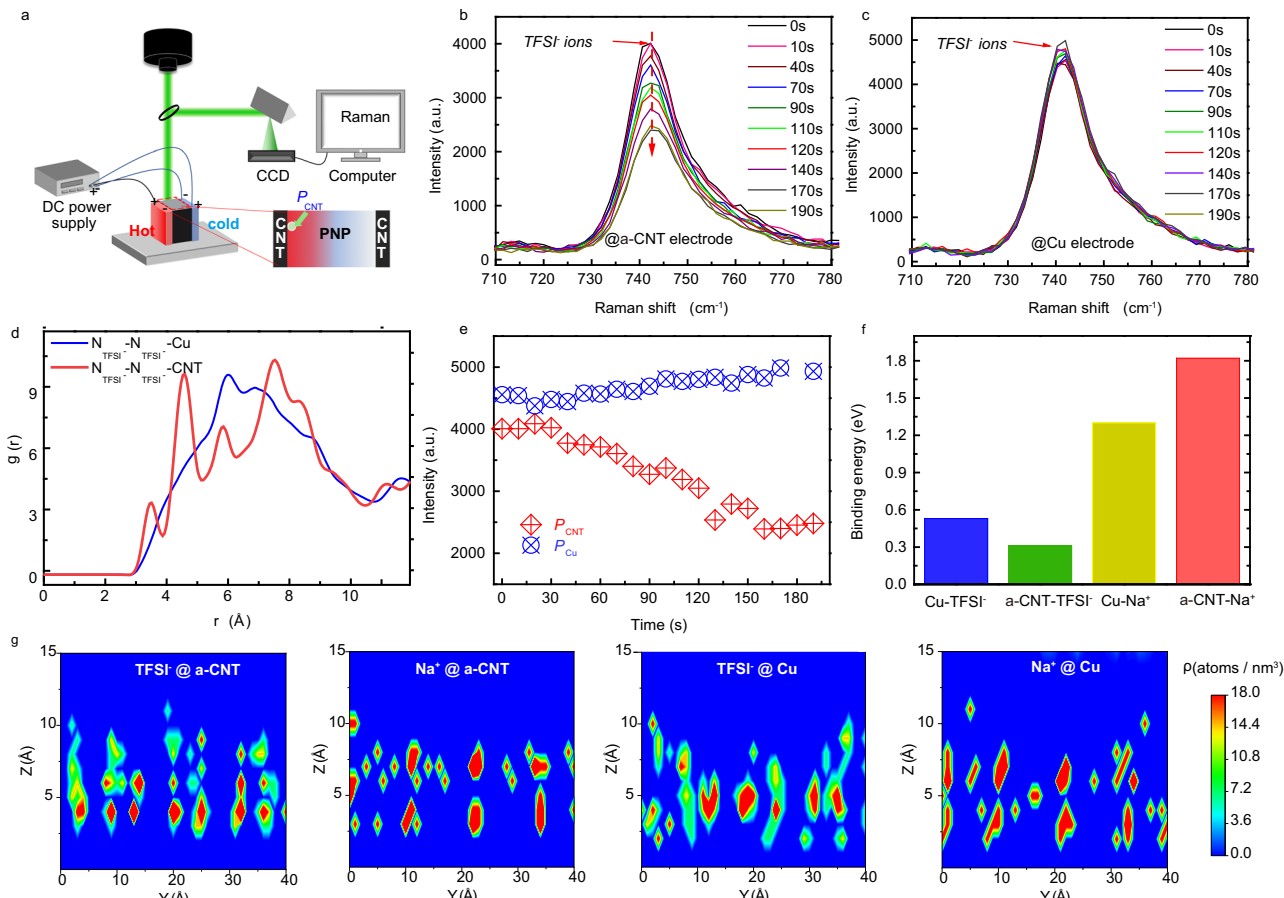

**Fig. 2 | Interfacial interaction behavior of Na+ and TFSI- ions on a-CNT and Cu electrode surfaces. a** The setup of the in-situ Raman measurement system. The magnitude variation of the TFSI- ions peak of **b** a-CNT|PNP|a-CNT and **c** Cu|PNP|Cu system in a range of 710–780 cm$^{-1}$. **d** The calculated radial distribution functions (RDF) of N$_{TFSI-}$–N$_{TFSI-}$ on a-CNT and Cu electrode surfaces. **e** The intensity variation of the TFSI- ions peak of a-CNT|PNP|a-CNT and Cu|PNP|Cu system at 742 cm$^{-1}$. **f** The interfacial interaction energy among (Cu, a-CNT) electrodes and (Na+, TFSI-) ions. **g** The 2D molecule number density diagram of TFSI- and Na+ ions on the a-CNT and Cu electrode surfaces.

Accordingly, we conducted a real-time in-situ Raman method to dynamically characterize the ions thermodiffusion process under a $\Delta T$. Intrinsically, the peak location and the intensity of the Raman spectrum can accurately detect the existence of ions and their concentration[28,29]. The PNP shows a predominate peak at 742 cm$^{-1}$, which is donated to the vibrational mode corresponding to the expansion and contraction of the whole TFSI- ions[30], whereas no such peak was observed from the mixture of PVDF-HFP and PC molecular, as shown in Fig. S8 (Supplementary Information). Accordingly, the Raman peak located at ~742 cm$^{-1}$ related to TFSI- anion is selected to investigate the electrode effect on the ion thermodiffusion process of the Cu|PNP|Cu, and a-CNT|PNP|a-CNT system.

As illustrated in Fig. 2a, the Cu|PNP|Cu or a-CNT|PNP|a-CNT systems were vertically exposed to the laser source when performed test and the $\Delta T$ was applied across PNP samples from the left (hot) to right (cold) side. The digital photos of the Cu|PNP|Cu and a-CNT|PNP|a-CNT systems are shown in Fig. S9a, b (Supplementary Information). The Raman line scans, consisting of ~20 scanning cycles (Fig. S9c, d, Supplementary Information), were performed every 10 s on a fixed location near the hot side interface of PNP/electrode ($P_{Cu}$ and $P_{a-CNT}$). It is observed that the magnitude of the Raman peak at ~742 cm$^{-1}$ corresponding to TFSI- ions at $P_{a-CNT}$ point varied slightly in the first 30 s (Fig. 2b, e). Interestingly, after 30 s, it is clear to find that the amplitude of the peak intensity of TFSI- ions significantly reduced until 160 s, which proves that the concentration of TFSI- ions at the $P_{a-CNT}$ point decreased. As the TFSI- ions near the interface of the hot side was

motivated by heat, these TFSI- ions started departing from the hot side to transport to the cold side. Accordingly, the TFSI- ions diffused away from the hot side toward the cold side and led to a decrease in the concentration of TFSI- ions near the hot interface after a certain period. The intensity of peak at ~742 cm$^{-1}$ reached a relatively steady state value of ~60% of the initial status. In contrast, for the Cu|PNP|Cu system, the variation of the magnitude of the TFSI- peak intensity at $P_{Cu}$ region is very limited (Fig. 2c, e), which is much more weakened than the a-CNT|PNP|a-CNT system. The Raman mapping strongly suggests that the TFSI- ion is more active near the a-CNT electrode than that of the Cu electrode, demonstrating the a-CNT electrode is more favorable for the diffusion of TFSI- ions.

Further, the surface morphology of Cu and a-CNT electrodes was studied by the scanning electron microscope (SEM). It is observed that the Cu electrode has a flat surface morphology, which is quite distinct from the rough surface of the a-CNT electrode in Fig. S10 (Supplementary Information). As known, the rough a-CNT electrode surfaces introduced a larger surface area and much more ions adsorption sites of PNP film compared with that of the flat Cu electrodes[31]. Accordingly, the molecular dynamics (MD) simulations were further performed to investigate the interfacial absorption behaviors of NaTFSI ($C_2F_6NNaO_4S_2$) with the a-CNT and Cu electrodes at the atomic scale (detail in Supplementary Methods). Fig. S11 (Supplementary Information) showed the top and side view of the MD snapshots of the Na+ cations and TFSI- anions on the Cu and a-CNT electrode surface at 300 K, respectively. The arrangement of TFSI- anions on electrode

surfaces were characterized by analyzing the radial distribution functions (RDF) of $N_{TFSI^-}$–$N_{TFSI^-}$ on these two electrode surfaces at 300 K, as shown in Fig. 2d. The RDF was defined as the probability of finding a nitrogen atom at a certain distance from another tagged nitrogen atom. The RDF of $N_{TFSI^-}$–$N_{TFSI^-}$ on a-CNT electrodes surface exhibits two obvious peaks at 4.7 and 7.3 Å (Fig. 2d), indicating the long-range order of TFSI anions polarization on the a-CNT surface, which is similar to the crystal-like structures[32]. By contrast, the two peaks of RDF at 6.1 and 7.6 Å on the Cu electrode surface are not well separated, indicating the TFSI anions are disorderly arranged near the Cu electrodes[33], which is also observed from the simulation of the mapping ions distribution near the electrode (Fig. 2g). The difference in the molecular arrangement of the TFSI anions are associated with the lattice match between the lattice structure of electrode surfaces and the TFSI anions, similar to the formation of ice crystal on the graphene oxide[34]. Moreover, it can be observed that the arrangement of TFSI anions are highly ordered near the surface of a-CNT electrodes compared with the relative disorder distribution of $Na^+$ ions on the a-CNT electrode surface (Fig. 2g). The ordered arrangement decreased the collision probability of TFSI anions, enhancing the thermodiffusion priority of TFSI anions near the a-CNT electrode surface compared to the $Na^+$ ions[35,36], which is consistent with the Raman experimental observations.

Meanwhile, the MD simulation found there is a significant difference in the number densities of both $Na^+$ cations and TFSI anions on the a-CNT and Cu electrode surfaces. On the a-CNT electrode surface, the peak position at 1.5 Å of the number density of $Na^+$ cations is much smaller than that of the peak at 4.7 Å of TFSI number density (Fig. S12a, Supplementary Information). By contrast, on the Cu electrode surfaces, the peak position (3.5 Å) of the number density of $Na^+$ ions is larger than that of TFSI ions (1.6 Å) (Fig. S12b, Supplementary Information). Such an obvious difference in the number density of ions is strongly related to the interaction strengths between electrodes and ions. The smaller peak position implied stronger interfacial interaction between ions and electrodes[37]. Hereby, it implied that the interfacial interactions between $Na^+$ ions and a-CNT electrodes are stronger than that of TFSI ions with a-CNT electrodes. To further quantitively study the interaction energy, the density functional theory (DFT) calculations of the interactions between the (a-CNT and Cu) electrodes and the (TFSI and $Na^+$) ions are performed[38,39] (see the "Methods" section and Fig. S13, Supplementary Information). From the DFT calculation results (Fig. 2f), it's interesting to find that the TFSI ions formed a stronger interaction with the Cu electrode (0.53 eV) than that interaction with the a-CNT electrode (0.31 eV). In contrast, the $Na^+$ ions formed a stronger interaction with the a-CNT electrode (1.82 eV) than the interaction strength with the Cu electrode (1.30 eV). The stronger interaction between $Na^+$ ions and the a-CNT electrodes could immobilize part of $Na^+$ ions and induce a larger drag force to $Na^+$ cations, impeding the departure process of $Na^+$ ions from a-CNT electrodes. As a result, the a-CNT|PNP|a-CNT exhibits the n-type characteristic.

Further, it is well known the temperature gradient provided the thermally driven force to motivate ions to depart from the hot side. But few reports have investigated the effect of the competition between the thermally driven force and the interfacial interaction induced by the electrode on the transport behaviors of ions. Hereby, a series of temperature differences (ΔT: 0−25 K) were applied to the a-CNT|PNP|a-CNT system to provide various strengths of thermal-driven force. At the original equilibrium status (ΔT = 0 K), the $Na^+$ cations and TFSI anions were evenly distributed in the material (Fig. 3a, stage I). Once the heating started, the $Na^+$ and TFSI ions of the a-CNT|PNP|a-CNT system were motivated, and the voltage between the hot and cold side ($\Delta V_{a-CNT}$) was produced correspondingly. As observed in Fig. 3b (stage II), the produced $\Delta V_{a-CNT}$ exhibited a positive value, which suggested a higher concentration of TFSI anions transported to the cold side (Fig. 3a, stage II), representing the *n*-type behavior of the a-CNT|PNP|a-

CNT system. When the ΔT was raised to 6 K, the amplitude of the produced $\Delta V_{a-CNT}$ was also increased. The amplitude of the $\Delta V_{a-CNT}$ gradually reached a saturated value when the ΔT reached near 12 K.

Surprisingly, the value of $\Delta V_{a-CNT}$ dramatically dropped, and the overall $\Delta V_{a-CNT}$ almost became neutral when the ΔT reached about 15 K (Fig. 3b, stage III). It implied that the number of effective $Na^+$ cations diffused from the hot side to the cold side is relatively equal to the amount of TFSI anions, which compromised the ionic Seebeck effect. It is suspected that the larger ΔT greatly strengthened the thermally driven force and enabled more amount of $Na^+$ cations to leave the hot a-CNT electrode, gradually balancing the $\Delta V_{a-CNT}$, as illustrated in Fig. 3a (stage III). Moreover, the $\Delta V_{a-CNT}$ was even changed to the negative direction under a large temperature difference (ΔT > 20 K) (Fig. 3c, stage IV). It is inferred that the gradually enhanced thermal-driven force on $Na^+$ ions is larger than the attraction force by the a-CNT electrode. As a result, a greater number of free $Na^+$ cations departed from the hot electrode $Na^+$ cations and dominated the thermodiffusion process (Fig. 3a, stage IV), which transformed from n-type to p-type characteristics of the a-CNT|PNP|a-CNT. Moreover, the thermoelectric voltage $\Delta V_{a-CNT}$ always exhibited a negative value during cyclic repeated testing at ΔT = 25 K, confirming the p-type behavior at a higher ΔT.

Clearly, the ions-electrode interfacial interaction and the thermally driven force significantly interfered with the thermodiffusion of $Na^+$ cations and TFSI anions. As shown in Fig. 3d, when the thermally driven force is lower than the ion−electrode interaction, the transportation of $Na^+$ cations will be restricted by the a-CNT electrodes and the a-CNT|PNP|a-CNT system performing n-type behavior. With the continued increasing ΔT, the thermally driven force was gradually enhanced. When the thermally driven force is stronger at a large ΔT, the restricted cations will depart from the hot side and diffuse to the cold side and the number of cations departing from the hot side could increase correspondingly. Until the amount of $Na^+$ cations moving to the cold end exceeds the number of TFSI anions, the $Na^+$ cations finally rule the thermodiffusion process, making the a-CNT|PNP|a-CNT system back to p-type. In this paradigm, adjusting the external electrode materials can effectively obtain a reversible bipolar thermopower to be capable of proceeding in either of two signs, which is significant to design advanced *i*-TE cells that generate cyclic energy under a constant heat source.

Additionally, the influence of asymmetric electrodes on the thermopower was further investigated. The PNP film was suspended on one a-CNT electrode and one Cu electrode as shown in Fig. S14 (Supplementary Information), which was donated as the a-CNT|PNP|Cu. The same PNP sample was used throughout the whole test, and the hot and cold sides were fixed to connect the work and counter electrodes of the voltage meter, respectively. Heating Cu side or a-CNT side, the positive sign of the ionic thermal voltages $\Delta V_{Cu-CNT}$ ($V_{Cu}–V_{a-CNT}$) or $\Delta V_{CNT-Cu}$ ($V_{a-CNT}–V_{Cu}$) were produced, and the magnitudes of generated voltage increased with enlarging the ΔT (regions I and II, Fig. S15, Supplementary Information), suggesting the TFSI ions dominated the thermodiffusion process (Fig. S16, Supplementary Information) and demonstrated n-type behaviors. Interestingly, once transferring the PNP from the hybrid Cu|a-CNT electrode to the symmetric Cu|Cu electrodes, it generated a negative sign of ionic thermal voltage $\Delta V_{Cu-Cu}$ ($V_{Cu}–V_{Cu}$) between the hot side and cold side in the region III (Fig. S15, Supplementary Information), behaving as p-type characteristic. In the following test (region IV, Fig. S15, Supplementary Information), this PNP film was transferred back to the a-CNT|Cu electrode and it performed n-type behavior again. Besides, it showed a high degree of reproducibility in response to the temperature differences after the cyclic test. The measured thermopower of the a-CNT|PNP|Cu system reached about −4.8 mV K$^{-1}$ (Fig. S17, Supplementary Information), which is around half of the a-CNT|PNP|a-CNT system. It may inspect that half part of the delocalized $Na^+$ cations at the Cu/PNP

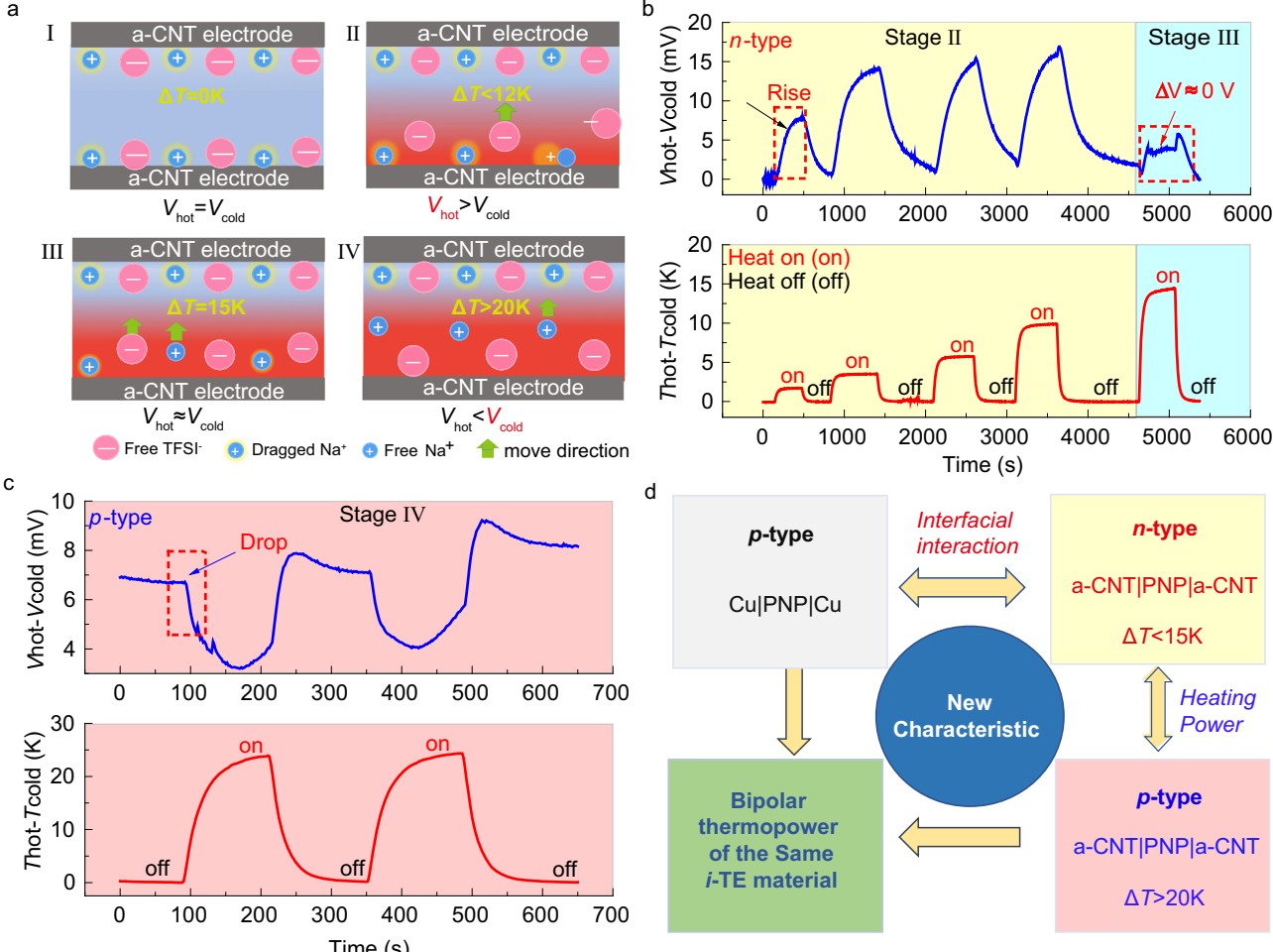

**Fig. 3 | The performance of the a-CNT|PNP|a-CNT system. a** The schematic of ion distribution of the a-CNT|PNP|a-CNT system at 4 stages (I) $\Delta T = 0$, (II) $\Delta T < 12$ K, (III) $\Delta T \approx 15$ K and (IV) $\Delta T > 20$ K. The curves of the generated thermoelectric voltage of the a-CNT|PNP|a-CNT system under various temperature differences of **b** $\Delta T < 15$ K and **c** $\Delta T > 20$ K. **d** The illustration of the mechanism of the bipolar thermopower characteristic of the PNP when using different electrodes and applying different heating power.

interface of the a-CNT|PNP|Cu system neutralized the thermoelectric voltage during the thermodiffusion process, compared to the a-CNT|PNP|a-CNT system. Clearly, this asymmetric electrode further confirmed that the manipulation of different electrode materials could lead to bipolar thermopower of the same *i*-TE PNP materials.

**Prototypes of i-TE generators and thermal sensors**

The previously reported *i*-TE supercapacitors[5,18,26] always require controlling the heat on and off process to push the back-and-forth movement of ions since the ions only accumulated at the electrode interface, which cannot achieve cyclic power generation under a constant heat source. To address such issues, we demonstrated a promising *i*-TE generator with an interesting reversible bipolar thermopower property that produces cyclic energy under a constant heat course that only requires switching the external electrodes. The *i*-TEG consists of 10 pairs of Cu electrodes and a-CNT electrodes which were connected in series to improve the output power, as illustrated in Figs. 4a and S18a (Supplementary Information). In brief, the mechanism is explained as followed. Firstly, the 10 pairs of the patterned Cu|Cu electrodes were tightly pressed to contact with the 10 pieces of PNP films from above (Fig. S18b, Supplementary Information). After inducing a temperature difference across 10 pieces of PNP films, taking the data in the first cycle in Fig. 4c as an example, it generated a negative thermal voltage ($V_i$) with Cu|Cu electrodes until reached a relatively stable value after heating as more amount of Na$^+$ cations moved to the

cold side (Fig. 4b-I). Next, the *i*-TEG was connected to a load to output power to the external circuit and the voltage $V_i$ dropped to near zero, which was caused by the accumulation of electrons and holes of the electrodes to balance $V_i$ (Fig. 4b-II). Then removing the Cu|Cu electrodes while keeping heating constant, the 10 pairs of the patterned a-CNT|a-CNT electrodes were switched to contact with the 10 pieces of PNP films (Fig. S18c, Supplementary Information) and the external resistor was disconnected simultaneously. It is clear to observe the *i*-TEG produced an opposite sign of thermal voltage (Fig. 4c). Because using a-CNT electrodes makes the PNP behave the n-type characteristic and the TFSI$^-$ anions dominated the thermodiffusion process (Fig. 4b-III). The last step was to connect the external resistor to the *i*-TEG again to produce the power to the external load (Fig. 4b-IV). Furthermore, after repeatedly alternatively switching the electrodes, the produced thermal voltage of the fabricated *i*-TEG demonstrated high repeatability after 20 cycles (Fig. 4c). Importantly, the proposed *i*-TEG achieves generating cyclic power under a constant heat source without the need to turn on/off the heat source or join/separate materials from the heat source. Interestingly, there is no necessity to change the thermal contact between the material and heat source and therefore the temperature difference does not need to be re-established, providing a significant innovative impact for expanding the practical applications.

To further explore the application of ionic thermoelectrics, a micro-solid-state ionic thermoelectric sensor was proposed to detect

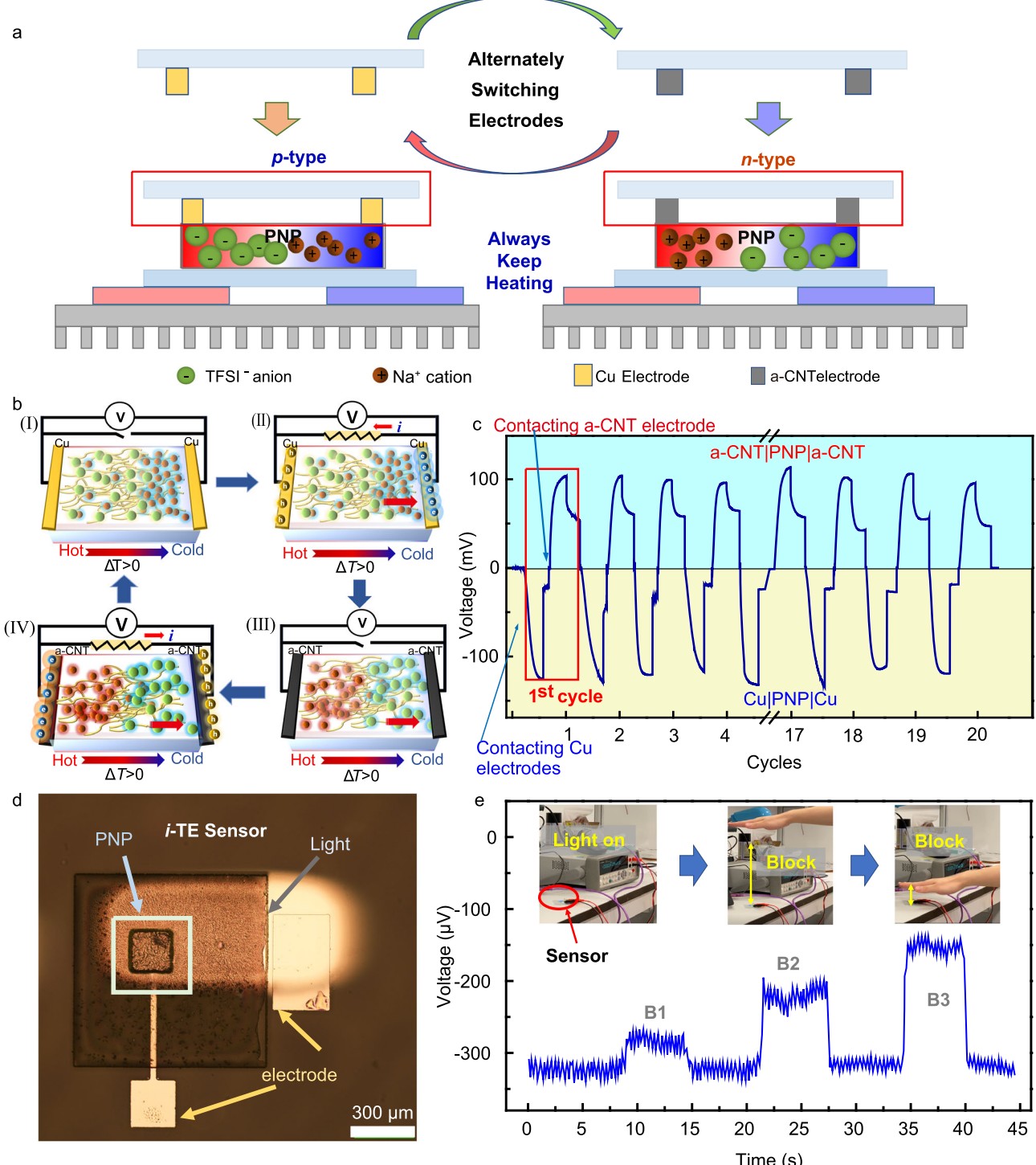

**Fig. 4 | The prototype of the fabricated ionic thermoelectric generator and sensor. a** The illustration of the working mechanism of the *i*-TEG with a constant Δ*T* by alternately switching the Cu|Cu, and a-CNT|a-CNT electrodes. **b** The illustration of the distribution of the Na⁺ cations and TFSI⁻ anions. **c** The working performance *i*-TEG produces cyclic power under a constant Δ*T*. **d** The digital image of the fabricated prototype of the *i*-TE sensor. **e** The performance of the fabricated *i*-TE sensor under different light illumination conditions.

light-induced temperature fluctuation, which is an essential part of the electronic skin network, thermal management, health monitoring, and other fields[40]. This micro *i*-TE sensor with a feature size of 200–300 μm (Fig. 4d) was fabricated by sandwiching the PNP film (*i*-TE layer) by a bottom electrode layer and a top metal/CNT electrode layer through the micro-electro-mechanical system (MEMS) technologies, which fabrication process flow was shown in Fig. S19 and Supplementary Methods (Supplementary Information). Briefly, the working

mechanism is to utilize the ionic thermoelectric to detect heat difference generated by the light, as illustrated in Fig. S20, (Supplementary Information). The *i*-TE sensor was initially exposed to an incandescent lamp for enough time and the produced voltages reached an equilibrium status. Once the light was blocked by hand, where the distance from the sensor is about B₁, it can be clearly observed the voltage signal was immediately changed (Fig. 4e). After removing the hand, the voltage quickly came back to its original status. Interestingly, when the

hand moved closer to the *i*-TE sensor with distances of B$_2$ and B$_3$, more amount of light was covered. Thus, it generated a larger $\Delta T$ between the top and bottom sides of the *i*-TE sensor. Accordingly, a much stronger voltage variation signal was observed (Fig. 4e and Supplementary Movie 1), demonstrating high sensitivity for detecting heat. The above results demonstrated that the promising capability for ions and electrode interactions can be used as an effective way of exploring unique properties and designing advanced ionic thermoelectric devices for energy generation and sensing.

## Discussion

This work reported a reversible bipolar thermopower behavior of the same *i*-TE PNP material by adjusting ion-electrode interactions and exhibited giant p-type (+20.2 mV K$^{-1}$) and n-type (−10.2 mV K$^{-1}$) thermopowers, respectively. The a-CNT formed stronger interactions with Na$^+$ ions and produced more impedance force to Na$^+$ cations, enabling TFSI$^-$ anions to govern the thermodiffusion process of the a-CNT|PNP| a-CNT system, performing n-type behavior. Moreover, the developed *i*-TEG successfully produced cyclic energy under a constant heat condition by utilizing the bipolar thermopower property, which is significant for expanding the applications of ionic thermoelectrics in the real world. In addition, the fabricated sensors performed high thermal sensibility and fast responsivity using the developed *i*-TE materials. This work provides significantly innovative impacts on exploring next-generation ionic thermoelectric materials and devices.

## Methods

### DFT details of interfacial interactions of ions with Cu and CNT electrodes

Density functional theory (DFT) calculations of the interactions between the electrodes (CNT and Cu) and the ions (TFSI$^-$ and Na$^+$) are performed by using the Vienna ab initio Simulation Package (VASP) code[38,39], where the exchange-correlation effects are treated by the generalized gradient approximation (GGA) in the Perdew–Burke–Ernzerhof (PBE) parametrization[41]. A vacuum layer >30 Å is set on the electrode surfaces. The TFSI$^-$ and Na$^+$ ions are separately put on the electrode surface and the structures are optimized. To embody the electronic states of ions, the total valence electronic numbers are set to be $N_{all-1}$ for the models with Na$^+$ ion and $N_{all+1}$ with TFSI$^-$ ion, where $N_{all}$ indicates the total of intrinsic valence electronic numbers as shown in Fig. S13 (Supplementary Information). In the geometry optimization, cutoff energy of 500 eV, energy convergence of 10$^{-5}$ eV, and force convergence of 10$^{-4}$ eV/Å are set. The total energy is recorded as $E_{tot,ads}$ Then, ions moved far away from the electrode by 15 Å, which is enough to exclude the interactions between electrodes and ions, and the total energy is recorded as $E_{tot,far}$ Lastly, adsorption energy, $E_{ad}$, can be calculated by $E_{ad} = E_{tot,\ far} - E_{tot,\ ads}$ A higher $E_{ad}$ means a stronger interaction between electrodes and ions.

## Data availability

The source data used in this study are available in the Figshare database (https://doi.org/10.6084/m9.figshare.21802899.v1) (ref. [42]). Extra data are available from the corresponding author from the corresponding authors (maweigang@tsinghua.edu.cn, W.G. Ma; mebhuang@ust.hk, B.L. Huang) upon reasonable request.

## Code availability

The code of the MD and DFT calculation that support the findings of this study are available within the article (refs. [38,39]) and its Supplementary Information file. Extra data are available from the corresponding author from the corresponding authors (maweigang@tsinghua.edu.cn, W.G. Ma; mebhuang@ust.hk, B.L. Huang) upon reasonable request.

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

## Acknowledgements

The authors are thankful for the financial support from the National Natural Science Foundation of China (Grant Nos. 52106095, 52176078, 52006130, 51827807, and 52250273), Science Fund for Creative Research Groups of the National Natural Science Foundation of China (Grant No. 51821004), the Tsinghua–Toyota Joint Research Fund. B.L.H. thanks the support from the Beijing Institute of Collaborative Innovation (Grant No. OKT20EG06), Hong Kong General Research Fund (Grant No. 16206020), and the Center on Smart Sensors and Environmental Technologies (Grant No. IOPCF21EG01) in the Hong Kong University of Science and Technology.

## Author contributions

C.C., W.G.M., and B.L.H. conceived and designed the project. W.G.M., B.L.H., X.Z.D., and X.Z. supervised this project. C.C. and G.Z.L. fabricated and tested the ionic thermoelectric materials and designed the experiment work. M.A., D.X.S., and Z.Q.W. contributed to the theoretical simulation. G.Z.L. and X.Q. performed the ionic Seebeck measurement. Y.F.Z. and C.Y.Z. performed the Raman test and discussion. Y.Z.D., Yang Lu, Z.Z.L., Yang Li, H.H., and C.J.L. contributed to the discussion of the results. C.C., M.A., and W.G.M. wrote the paper. All authors contributed to the discussion and manuscript preparation.

## Competing interests

The authors declare no competing interests.
