## [Peer Review File · Nature Communications]

Reversible Bipolar Thermopower of Ionic Thermoelectric Polymer Composite for Cyclic Energy GenerationREVIEWER COMMENTS

Reviewer #1 (Remarks to the Author):

The results presented in this manuscript are interesting. However, some issues should be understood better. (1) As Cu is an active metal, the authors should check whether there is any reaction at the interface between the polymer composite and the Cu electrodes. (2) The authors should carry out control experiments by using stable metals like Au or Pt as the electrodes. (3) As the authors mentioned, the interaction between the CNT electrodes and the polymer composites can greatly affect the thermovoltage. Details should be provided on the fabrication of VA-CNTs, the structure like the length and diameter and properties like the conductivity of the VA-CNTs, and the fabrication procedure of the CNT electrodes. (4) The contact area between the CNT electrodes and the polymer composites can affect their interaction and thus the thermovoltage. The authors should study the effect of the contact area on the thermovoltage, such as using VA-CNTs of different heights along the vertical direction. (5) Can the n-type behaviour be observed by using single-walled CNTs or multi-walled CNTs commercially available? (6) For each measurement, please also check the thermovoltage values under negative temperature gradients in addition to the positive gradients. (7) Understand that the samples were prepared and they were tested in inert environment. Does the humidity affect the thermovoltages using VA-CNTs or Cu as the electrodes, because humidity usually affect the ionic thermovoltage.

Reviewer #2 (Remarks to the Author):

This is an interesting paper on ionic thermoelectric materials and devices. As their main result the authors present a novel technique of transforming waste heat to electric energy.

In previous work this was achieved by periodically disconnecting the iTE material from the heat source, the present work realizes a cyclic thermopotential by periodically connecting two types of electrodes. As their essential feature, these electrodes modify the TE response and in particular the sign of the Seebeck coefficient.

Though this is possibly a very interesting paper I do not recommend publication in its present form. Several points need to be clarified:

1) For applications the most important aspect is change of S after switching from one electrode to another, as illustrated by regions I-IV in Fig S6 of the supplemental material. The switching process is shown schematically in fig. 4a.

In the introduction (p 3) the authors say that "... it is impossible to repeatedly turn on/off the heat in the real industry..." It is not obvious to me why it would be easier to change electrodes. Perhaps the authors could try to expand on their argument.

2) If I am not mistaken the authors observe $S = 20$ mV/K when using Cu electrodes, and $S = -10$ mV/K for CNT electrodes. This finding cannot be explained in terms of ionic heats of transport Q^+ and Q^- , since the latter are material properties, which by definition are independent of the electrodes. The MD simulations suggest that the electrodes modify the ionic probability distribution in a nanoscale surface layer. Since this is the main finding of their paper, the authors should try to explain or at least better characterize experimentally this effect.

3) I feel there is a misunderstanding concerning eq. (1) and the subsequent discussion. In fact, eq. (1) does not give the stationary Seebeck coefficient of iTE materials, but rather the transient ion current after switching on the temperature gradient.

Thus eq. (1) would explain the initial slope of the cusps occurring in Fig. 1 d,e after each change of ΔT . Perhaps the authors find it helpful to have a look at ref. 18 or at a paper by Janssen and Bier (Phys. Rev. E 99, 2019).

Reviewer #3 (Remarks to the Author):

The author used the interaction between ions and electrodes to control the migration of cations and anions by changing the electrode materials, and achieved positive and negative conversion of the thermopower of PNP films. This reversible p-n transition provides a new approach for the construction of thermodiffusion ionic thermoelectric devices. However, this work is incomplete, and there are many issues which should be further clarified. I also have some suggestions for the authors to further polish their paper.

1. The title of this paper says “ionic thermoelectric polymer composite for continuous energy generation”, however, the work on ionic thermoelectric generator is not yet complete. The low power generation of a single PNP film is not meaningful to evaluate its performance. The author needs to connect multiple PNPs in series and consider its stability after many cycles.
2. According to Fig. 4, it needs switching the electrodes while also disconnecting the external load and waiting for a certain period to generate the open circuit voltage. I think the generator does not produce power continuously, which is not consistent with the “continuous power generation” claimed in the title of this paper. In addition, there is an error in the labeling sequence of Fig. 4b.
3. For the measurement of the Seebeck coefficient, the connection between the positive and negative poles of the voltmeter and the hot and cold ends of materials will affect the positive and negative values of voltages. Therefore, please supply the measurement details, schematic and physical diagram of measurement equipment for the Seebeck coefficient. Moreover, in Fig. 1d, the relationship between the thermoelectric voltage and temperature difference of p-type Cu|PNP|Cu does not correspond to Fig. S2. The same problem exists in Fig. 1e.
4. What are the advantages of VA-CNT as electrodes compared to other carbon materials such as SWCNT or MWCNT? The reason for choosing VA-CNT arrays as electrodes should be described.
5. There is an error in the figure caption of Fig. 2a and Fig. 2b, which is inconsistent with the description in line 185 on page 10.
6. The ionic thermoelectric polymer was described as all-solid-state i-TE material. However, this i-TE material contains liquid propylene carbonate (PC), so it is not considered to be an all-solid material.

Revision Report

Response to the comments by the referees on the manuscript (NCOMMS-22-27100-T) entitled “by C. Chi[†], G. Liu[†], M. An[†], Y. Zhang, D. Song, X. Qi, C. Zhao, Z. Wang, Y. Du, Z. Lin, Y. Lu, H. Huang, Y. Li, C. Lin, W. Ma*, B. Huang*, X. Du, and X. Zhang”. We appreciate the editor and the referees for their careful consideration, insightful comments, constructive suggestions, and positive evaluation of improving our manuscript. Accordingly, we have carefully revised our manuscript according to the referees’ suggestions. All the revised and added parts have been marked in red color in the manuscript. Below are the point-by-point responses to the referee’s comments.

Comments from the editors and referees:

Reviewer #1 (Remarks to the Author):

The results presented in this manuscript are interesting. However, some issues should be understood better.

(1) *As Cu is an active metal, the authors should check whether there is any reaction at the interface between the polymer composite and the Cu electrodes.*

Response: As the reviewer suggested, we further conducted to study on the stability of Cu electrodes when contacting the polymer composite during the thermopower measurement. In this work, the Cu electrodes used for the test are conductive copper tapes. Firstly, we checked the surface color of the Cu electrodes before and after the thermopower measurement, and don’t observe any color change on the surface of the used Cu electrodes. We further performed the X-ray photoelectron spectroscopy (XPS) measurement to characterize the chemical state of the surface of the unused Cu

electrode and the used Cu electrode after the thermopower testing of the PNP composite. As shown in Fig. R1a, the XPS curves of the unused Cu and the used Cu electrodes overlapped well and no new peaks were observed on the used Cu electrodes, which suggests that there is no chemical reaction on the Cu electrode during the test. Moreover, we also investigated the cycle stability of the PNP *i*-TE polymer materials using the Cu electrodes. As shown in Fig. R1b, there is no obvious decay of the produced thermoelectric voltage of the Cu|PNP|Cu system after 30 cycles test. Thus, the above test results represented the high stability of the Cu electrodes with the PNP composite. In addition, the Cu electrode is also widely used to work as a test electrode when measuring the ionic Seebeck coefficient of polymer composite in some recent work (*Adv. Energy Mater.* 2019, 9, 1901085).

We have added the discussion of the stability of Cu electrodes in the revised manuscripts (Page #6, Line 115) and Supplementary Information (Page #7, Fig. S4).

Fig. R1 The XPS characterization of the surface of the Cu electrode before and after thermopower measurement. (b) The cycle stability test of the PNP *i*-TE materials with Cu electrodes.

(2) The authors should carry out control experiments by using stable metals like Au or Pt as the electrodes.

Response: As the reviewer suggested, we conducted the control experiment using stable metals, Pt and Au as the electrodes. To keep consistent, the thermopower measurement setups of the Pt|PNP|Pt and Au|PNP|Au systems are the same as the

Cu|PNP|Cu system, at 298K and 70% relative humidity (RH). Different intensities of temperature differences ($\pm 6\text{K}$ to $\pm 1\text{K}$) were applied across the Pt|PNP|Pt and Au|PNP|Au systems. Figs. R2a and 2b show the measured thermopower curves of the PNP *i*-TE films tested by Pt and Au electrodes. The ionic Seebeck coefficient (S_i) of *i*-TE material is defined as Eq. R1

$$S_i = -\frac{V(T_H) - V(T_C)}{T_H - T_C} \quad (\text{R1})$$

where $V(T_H)$ and $V(T_C)$ correspond to the voltage of the hot electrode at temperature T_H and the cold electrode at temperature T_C , respectively, (*Sci. Adv.* **7**, eabi7233 2021; *J. Chem. Phys.* **154**, 164511, 2021). The measured ionic Seebeck coefficient of Pt|PNP|Pt, and Au|PNP|Au are $19.3 \pm 3.2 \text{ mV K}^{-1}$ and $21.1 \pm 3.8 \text{ mV K}^{-1}$ (Fig. R2c), respectively. From the test results, both Pt|PNP|Pt and Au|PNP|Au systems also exhibited typical *p*-type characteristics, which is consistent with the value of thermopower ($\sim 20 \text{ mV K}^{-1}$) measured by the Cu electrodes.

We have added the discussion of the control experiments by using stable metals, Au and Pt, as the electrodes to the revised manuscript (Page #6, Line 121) and Supplementary Information (Fig. S3).

Fig. R2 The measured thermal voltage curves of (a) Pt|PNP|Pt system and (b) Au|PNP|Au system as a function of time under various $\pm \Delta T$. (c) The fitting curves of the $-(V_{hot} - V_{cold})/\Delta T$ of the Pt|PNP|Pt, and Au|PNP|Au systems.

(3) *As the authors mentioned, the interaction between the CNT electrodes and the polymer composites can greatly affect the thermovoltage. Details should be provided on the fabrication of VA CNTs, the structure like the length and diameter and properties like the conductivity of the VA CNTs, and the fabrication procedure of the CNT electrodes.*

Response: As suggested by the reviewer, we further introduce the properties of the CNT electrodes. The aligned CNT films were purchased from Nanjing Ji Cang Nano Technology Co., Ltd. (Nanjing, China), and the product number is JCNTF-20C, which is a kind of CNT-based thin film. As provided by the vendor, the aligned CNT film was fabricated by the chemical vapor deposition (CVD) method and continuously grown into a large-scale thin film of about 100×150 cm and has a very thin thickness of about ~10 μm, as shown in Fig. R3a. The surface morphology of aligned CNTs was also characterized by SEM and TEM. As shown in Fig. R3b, the aligned CNT consists of bundles of carbon tubes which were mainly aligned in the length direction of the film with small entanglements. This aligned-CNT exhibits high conductivity of $3 \times 10^5 \text{ S m}^{-1}$, low sheet resistance of $1 \sim 1.5 \text{ } \Omega \text{ m}^{-2}$, high mechanical strength of ~120 MPa, high stability and flexibility, low cost, and self-adhesive nature.

As the reviewer suggested, the detail of the properties of the aligned CNTs has been updated in the revised manuscript (Page #7, Line 143) and Supplementary Information (Page #S25, Fig. S5).

Fig. R3 (a) The digital photo and (b) the SEM image (with inset: the TEM) of the

aligned-CNT film.

(4) The contact area between the CNT electrodes and the polymer composites can affect their interaction and thus the thermovoltage. The authors should study the effect of the contact area on the thermovoltage, such as using VA CNTs of different heights along the vertical direction.

Response: Yes, we think the name of the vertical-aligned CNTs may lead to misunderstanding for the reviewer. In this work, the “aligned CNT” is a kind of carbon nanotube thin film with an average thickness of $\sim 10\ \mu\text{m}$, which is made of bundles of carbon tubes arranged uniformly in the length direction (Fig. R3b). **When testing the thermopower, the aligned CNT films were placed horizontally, and the length direction of the aligned CNT was parallel to the *i*-TE polymer material, which is not vertical to the polymer material.** As the a-CNT film is very thin and easy to bend when vertically placed, it is not suitable for testing thermopower by placing this a-CNT vertically to the PNP *i*-TE materials. **To avoid misunderstanding, we have changed the name “vertical aligned CNT” to “aligned CNT (a-CNT)” in the revised manuscripts.**

Moreover, as the reviewer suggested, we further investigated the effect of the electrode thickness on the generated thermal voltage. The a-CNT films of different thicknesses (T_a : $10\ \mu\text{m}$, T_b : $20\ \mu\text{m}$, and T_c : $30\ \mu\text{m}$) provided by the same vendor (Fig. R4) were cut into the same size ($1.0\ \text{mm} \times 20\ \text{mm}$) to be employed as the tested electrodes when performing the measurement. And *i*-TE PNP films were cut into identical rectangular shapes from one matrix and suspended on each a-CNT|a-CNT electrode. As shown in Figs. R4b-d, after introducing the same ΔT , the amplitude of produced voltages at every ΔT of three electrodes are almost equal, suggesting the thermopower is independent of the thickness of the a-CNT electrodes. In addition, to further study the effect of surface contact area, another aligned CNT film (named a-CNT-B) with a similar thickness of $8\sim 10\ \mu\text{m}$ but a bit lower specific surface area (SSA) was purchased from a different vendor (Chengdu Jiakai Technology Co., Ltd.). The tested thermopower is about $-8.7 \pm 1.05\ \text{mV K}^{-1}$ using the a-CNT-B electrodes under

the same test condition, which is a bit lower than the a-CNT|PNP|a-CNT (-10.2 ± 0.83 mV K⁻¹), as shown in Fig. R4e. It's possible that the differences may be originated from different preparation procedures and quality control by different vendors and a relatively smaller SSA.

Thus, the sign of the thermopower is independent of the contact area of the aligned CNTs, but different SSA of a-CNTs may affect the magnitude of the produced voltages, as a larger specific surface area usually store more amount of ions near the interface (*Nat. Mater.* 2008;7:845–854; *Nat. Energy.* 2016;1:16070).

Fig. R4 (a) The digital photos of the a-CNT with different thicknesses (T_a : 10 μm, T_b : 20 μm, and T_c : 30 μm). The measured thermoelectric voltages of PNP with the three a-CNT electrodes with different thicknesses of (b) 10 μm, (c) 20 μm, and (d) 30 μm. (e) The fitting curves $-\Delta V$ vs ΔT of the PNP with a-CNT and a-CNT-B (another vendor) electrodes.

(5) *Can the n-type behaviour be observed by using single-walled CNTs or multi-walled CNTs commercially available?*

Response: Yes, the *n*-type behavior is also observed by using single-walled CNTs or multi-walled CNTs as the test electrodes. As the reviewer suggested, we also conducted to measure the thermopower of PNP using the single-walled CNTs (SWCNTs) and multi-walled CNTs (MWCNTs) as electrodes, where the digital photos of the a-CNTs, MWCNT, and SWCNT are shown in Figs. R5a-c. Same to the thermopower measurement setup of the a-CNT|PNP|a-CNT system, the measured thermopower of PNP tested by MWCNTs and SWCNTs are $-8.17 \pm 1.2 \text{ mV K}^{-1}$ and $-5.2 \pm 1.35 \text{ mV K}^{-1}$, respectively, which is relatively lower than that of $-10.2 \pm 0.83 \text{ mV K}^{-1}$ of a-CNTs. And the MWCNT|PNP|MWCNT and SWCNT|PNP|SWCNT systems also perform *n*-type characteristics, as shown in Figs. R5d-e. Due to the softness and self-adhesive nature of the a-CNT material, it is very easy to contact tightly with the PNP polymer compared to the thick SWCNTs and MWCNTs films. Having good contact is of great significance for reducing contact resistance and improving the overall ionic thermoelectric conversion performance. Moreover, since the bundles of CNTs are well aligned along the length direction with small entanglements, a-CNT films exhibited higher conductivity of $3 \times 10^5 \text{ S m}^{-1}$, a lower sheet resistance of $1 \sim 1.5 \text{ } \Omega \text{ m}^{-2}$, excellent flexibility, higher mechanical strength of $\sim 120 \text{ MPa}$ and low cost compared to common SWCNT and MWCNT films, as summarized in Table R1.

We have added the thermopower performance of the SWCNT and MWCNT electrodes in the revised manuscript (Page #7, Line139, Figs. 1d-e) marked with red color and Supplementary Information (Page #S8, Fig. S5 and Page #S25, Table S1).

Table R1 The comparison of the physical property of the three types of CNTs.

Property	Thickness	Conductivity	Sheet resistance	Strength	Length	Diameter
----------	-----------	--------------	------------------	----------	--------	----------

aligned CNT	$\sim 10 \mu\text{m}$	$\sim 3 \times 10^5 \text{ S m}^{-1}$	$1 \sim 1.5 \Omega \text{ m}^{-2}$	$\sim 120 \text{ MPa}$	continuous growth	20-50 nm
MWCNT	$\sim 0.15 \text{ mm} \pm 0.05 \text{ mm}$	/	$\sim 4 \Omega \text{ m}^{-2}$	$\sim 10 \text{ MPa}$	10-30 μm	20-30 nm
SWCNT	60-80 μm	$2 \sim 4 \times 10^3 \text{ S m}^{-1}$	/	10-15 MPa	10-30 μm	1-2 nm

Fig. R5. The digital and SEM images of (a) aligned CNT, (b) MWCNT, and (c) SWCNT electrodes. (d) The fitting curves of the measured thermal voltage under different temperature gradients and (e) the corresponding thermopower of a-CNT|PNP|a-CNT, MWCNT|PNP|MWCNT, and SWCNT|PNP|SWCNT systems.

(6) For each measurement, please also check the thermovoltage values under negative temperature gradients in addition to the positive gradients.

Response: As the reviewer suggested, we further tested the thermopower of Cu|PNP|Cu, and a-CNT|PNP|a-CNT systems by introducing temperatures in both positive and negative temperature gradients ($\pm\Delta T_1$) (i.e., +6K, -6K, +5K, -5k... and ± 1 K). The re-measured generated voltage (V_1) of the Cu|PNP|Cu system under different ΔT_1 is shown in Fig. R6a, which exhibited a typical thermoelectric behavior. As shown in Fig. R6c, the slope of the fitting curve of re-measured $-\Delta V_1$ ($V_{hot}-V_{cold}$) vs ΔT_1 is close to the previous data acquired under positive temperature gradients ($-\Delta V_2$ vs ΔT_2) in the first submitted manuscript. According to Eq. R1, the re-measured thermopower of Cu|PNP|Cu is about 20.2 ± 4 mV K⁻¹, which is consistent with the previously measured results (~ 20 mV K⁻¹), as shown in Fig. R6c. Meanwhile, by applying the positive and negative temperature gradient (ΔT_3) to the a-CNT|PNP|a-CNT system, the re-measured thermopower of PNP is -10.2 ± 0.83 mV K⁻¹ by fitting the slope of $-\Delta V_3$ vs ΔT_3 , which is very close to the previously tested thermopower of -10.64 mV K⁻¹ (slope of $-\Delta V_4$ vs ΔT_4) under only positive temperature difference (ΔT_4), as shown in Figs. R6b and 6d. Clearly, the measured thermopower of PNP using positive and negative temperature differences is consistent with the measured results by using the single positive temperature difference.

The latest measured thermopower using both positive and negative temperature gradients has been added in the revised manuscript marked with red color (Page #5, Line 107) and in Supplementary Information (Page #S5, Fig. S2).

Fig. R6 (a) The re-measured voltage V_1 of Cu|PNP|Cu under positive and negative temperature difference ΔT_1 (red line) and (b) the re-measured voltage V_2 of a-CNT|PNP|a-CNT under positive and negative temperature difference ΔT_2 (red line). (c) The comparison of the fitting curves of the re-measured ($-\Delta V_1$ vs ΔT_1) with the previous data ($-\Delta V_2$ vs ΔT_2) of Cu|PNP|Cu samples. (d) The comparison of the fitting curves of the re-measured ($-\Delta V_3$ vs ΔT_3) under positive and negative temperature difference with the previous data ($-\Delta V_4$ vs ΔT_4) of a-CNT|PNP|a-CNT samples under positive temperature difference.

(7) Understand that the samples were prepared and they were tested in inert environment. Does the humidity affect the thermovoltages using VA-CNTs or Cu as the

electrodes, because humidity usually affect the ionic thermovoltage.

Response: According to the reviewer's suggestion, the thermopowers of the PNP samples using Cu, and a-CNT electrodes at different relative humidity (50% RH~90% RH) were further investigated. The curves of the produced thermal voltage of Cu|PNP|Cu sample at 90% RH was shown in Fig. R7a and the fitting curves of $-\Delta V/\Delta T$ at different humidity conditions under various temperature differences ($\Delta T = \pm 1$ K to ± 7 K) were demonstrated in Fig. R7b. The ionic thermopower of Cu|PNP|Cu increased from 8.12 mV K^{-1} (50% RH) to 26.62 mV K^{-1} (90% RH), as shown in Fig. R7c. According to previous work (*ACS Appl. Polym. Mater.* 2019, 1, 2723-2730; *Adv. Energy Mater.*, 2015, 5, 1500044 and *Adv. Mat. Res.* 2012, 501, 39-43), the porous structure of PVDF-HFP and the hydrophilic nature of sodium salt tend to absorb water from the moisture environment and could fill the space in the PVDF-HFP matrix. The absorbed water could help improve the dissociation of the NaTFSI ion pairs by weakening the electrostatic attraction. Moreover, further increasing the humidity level may create continuous water percolation paths or ion transport channels, which could reach a saturation state.

Fig. R7 The thermopower performance of Cu|PNP|Cu sample under different relative humidity. (a) The measured thermoelectric voltage at various temperature differences

of Cu|PNP|Cu sample at RH 90%, (b) the plot of $-(\Delta V)-\Delta T$ fitting curves, and (c) the thermopower of Cu|PNP|Cu sample various humidity ranging from RH 50% to RH 90%.

Meanwhile, the *i*-TE performance of the *n*-type CNT|PNP|CNT system was also investigated under different humidity. The measured thermal voltage of the a-CNT|PNP|a-CNT sample under different ΔT at RH 70% is shown in Fig. R8a. The ionic thermopower of the *n*-type a-CNT|PNP|a-CNT sample slightly increased from $-8.11 \pm 1.5 \text{ mV K}^{-1}$ at RH 50% to $-9.3 \pm 0.7 \text{ mV K}^{-1}$ at RH 60% and reached the maximum value of $-10.2 \pm 0.83 \text{ mV K}^{-1}$ at RH 70%, which was caused by water absorption effect as discussed above. Further increasing the humidity level, the thermopower reached a relative saturation value of $-9.7 \pm 0.86 \text{ mV K}^{-1}$ at RH 80% (Figs. R8b-c).

In summary, increasing the humidity level generally enlarged the magnitude of the thermopower of *p*-type Cu|PNP|Cu and *n*-type CNT|PNP|CNT, but it does not influence the sign of the *p*-type characteristic of the Cu|PNP|Cu and the *n*-type behavior with a-CNT|PNP|a-CNT. We have added the discussion of the humidity effect on the thermopower in the revised manuscript (Page #8, Line 161) marked with red color and Supplementary Information (Page #S9, Fig. S6 and Page #S10, Fig. S7).

Fig. R8. The thermopower performance of a-CNT|PNP|a-CNT sample under different relative humidity. (a) The measured thermoelectric voltage at a series of temperature differences of a-CNT|PNP|a-CNT sample at RH 70%. (b) The plot of $-(\Delta V)/\Delta T$ fitting curves and (c) the measured thermopower of a-CNT|PNP|a-CNT sample at various humidity ranging from RH 50% to RH 80%.

Reviewer #2 (Remarks to the Author):

This is an interesting paper on ionic thermoelectric materials and devices. As their main result the authors present a novel technique of transforming waste heat to electric energy. In previous work this was achieved by periodically disconnecting the iTE material from the heat source, the present work realizes a cyclic thermopotential by periodically connecting two types of electrodes. As their essential feature, these electrodes modify the TE response and in particular the sign of the Seebeck coefficient. Though this is possibly a very interesting paper I do not recommend publication in its present form. Several points need to be

clarified:

(1) For applications the most important aspect is change of S after switching from one electrode to another, as illustrated by regions I-IV in Fig S6 of the supplemental material. The switching process is shown schematically in fig. 4a. In the introduction (p 3) the authors say that "... it is impossible to repeatedly turn on/off the heat in the real industry..." It is not obvious to me why it would be easier to change electrodes. Perhaps the authors could try to expand on their argument.

Response: Yes, as the reviewer suggested, we further expanded the discussion on the advantages of the *i*-TE generators (*i*-TEGs) based on the unique bipolar thermopower property (*p*- and *n*-type) as followed.

For the ionic thermoelectric conversion based on the Soret effect, the ions cannot directly transfer across the electrodes but accumulate at the electrode interface under a temperature gradient. As a result, the fabricated *i*-TEGs operated in a capacitive charge/discharge mode. They must alternatively control the on/off status of the heat source to push ions transport forward and backward. Or the *i*-TE materials must be alternately attached and separated from the heat source to ensure the repeated movement of ions to achieve thermoelectric conversion. Unfortunately, the above method requires changing the thermal contact between the heat source and the *i*-TEG, and the thermal gradient needs to be re-established on the *i*-TE material, which cannot achieve cyclic power generation under a constant heat source, resulting in reduced conversion efficiency in practical applications. (*Energy Environ. Sci.* **9**, 2016, 1450-1457; *Nat. Mater.* **18**, 608-613, 2019; *Adv. Energy Mater.* **9**, 2019, 1901085; *Nat. Commun.* **10**, 2019, 1093).

To address the above issues, **this work proposed a facile way to achieve a cyclic conversion under a constant heat source by a new operation principle that only requires switching the external electrodes. Especially, the *i*-TE PNP materials are always contacting a constant heat source, without needing to turn on/off the heat source or attach/detach the materials from heat sources. More importantly, there**

is not necessary to change the thermal contact between the material and heat source, and thus don't need to re-establish the temperature difference either, which can favor improving the efficiency. In addition, to better demonstrate the advantages of the interesting properties, we further optimized the structure of the ionic thermoelectric generator (*i*-TEG) module. As shown in Fig. R9, the *i*-TE materials were always kept in contact with the Peltier heater and colder located at the bottom. When the Cu and a-CNT electrodes were alternately contacted with PNP directly from the top direction, the PNP *i*-TE materials exhibited *p*-type or *n*-type behaviors, respectively. As a result, the *i*-TEG achieves cyclically converting heat to power by alternatively switching a-CNT and Cu electrodes under a constant temperature difference. We believe the proposed work mechanism is more convenient and efficient for practical applications compared to the previously reported methods.

Meanwhile, as the reviewer commented, *we have deleted the content "it is impossible to repeatedly turn on/off the heat in the real industry" and further expanded the argument on the advantages of changing electrodes in the introduction part of the manuscript, as followed. "The heat source must be repeatedly established and removed for every charging and discharging cycle to ensure ions move back and forth, which is not convenient in practical applications. Although the i-TE devices periodically contacted the heat and cold source providing another way to convert heat to power, the i-TEGs still need to detach from the heat sources and take time to re-establish the temperature difference in every cycle, which increases energy consumption and reduce the conversion efficiency. Further efforts are needed to address the limitations of ionic thermoelectric technology applications."* (Manuscript, Page #3, Line 59).

Fig. R9. The illustration of the working principle of the proposed *i*-TEG converts heat to power under a constant heat source.

(2) *If I am not mistaken the authors observe $S = 20 \text{ mV/K}$ when using Cu electrodes, and $S = -10 \text{ mV/K}$ for CNT electrodes. This finding cannot be explained in terms of ionic heats of transport Q^+ and Q^- , since the latter are material properties, which by definition are independent of the electrodes. The MD simulations suggest that the electrodes modify the ionic probability distribution in a nanoscale surface layer. Since this is the main finding of their paper, the authors should try to explain or at least better characterize experimentally this effect.*

Response: This work reported a reversible bipolar thermopower ($+20 \text{ mV K}^{-1}$ to -10 mV K^{-1}) of the same PNP by testing with Cu and a-CNT electrodes. The MD simulation study found the distribution of TFSI ions near the a-CNT surface is similar to the crystal-like structure (favor TFSI⁻ ions to diffuse) and a-CNT have stronger interaction with Na⁺ ions (inhibit Na⁺ ions to diffuse), resulting in *n*-type behavior of a-CNT|PNP|a-CNT. **As the reviewer suggested, we further conducted experiments study of the electrode effect on ion distribution by real-time *in-situ* Raman method, which can dynamically characterize the ions thermodiffusion process under a temperature gradient. And density functional theory (DFT) was conducted to quantitatively investigate the interaction strength between Na⁺ ions, TFSI⁻ ions and Cu, a-CNT electrodes.**

In-situ Raman microspectroscopy is a powerful tool to study the ion transport properties of a material. Especially, the peak location and the intensity of the Raman spectrum can accurately detect the existence of ions and their concentration as suggested by the previous research (*Nat. Commun.* 12, 4053, 2021; *Nat. Commun.* 13, 5330, 2022; *J. Power Sources*, 449, 227361, 2020). Accordingly, the Raman mapping method is utilized for dynamically monitoring the transportation process of ions under a given temperature gradient in real time. In this study, the Raman spectra were captured by the HORIBA LabRAM HR Evolution Raman system with a 633 nm excitation laser with a light spot size of about 0.34 μm . And the Raman measurement was calibrated using standard Si materials. For the control experiment, it is clear to find that the PNP shows a predominate peak at 742 cm^{-1} , which is donated to the vibrational mode corresponding to the expansion and contraction of the whole TFSI⁻ ions (*J. Power Source*, 245, 630-636, 2014), whereas no such peak was observed from the mixture of PVDF-HFP and PC molecular, as shown in Fig. R10. Accordingly, the Raman peak located at $\sim 742\text{ cm}^{-1}$ related to TFSI⁻ anion is selected to investigate the electrode effect on the ion thermodiffusion process of the Cu|PNP|Cu, and a-CNT|PNP|a-CNT system.

Fig. R10 The Raman spectra of (a) PNP, (b) PVDF-HFP and (c) NaTFSI/PC.

Fig. R11 (a) The setup of the Raman measurement system. The images of the cross-section view of the (b) CNT|PNP|CNT and (c) Cu|PNP|Cu system.

As illustrated in Fig. R11a, the Cu|PNP|Cu and a-CNT|PNP|a-CNT systems were vertically exposed to the laser source when performed test and the temperature gradient was applied across PNP samples from the left (hot) to right (cold) side. The digital photos of the Cu|PNP|Cu and a-CNT|PNP|a-CNT systems are shown in Figs. R11b and c. The Raman line scans, consisting of ~ 20 scanning cycles, are performed every 10s on a fixed location near the hot side interface of PNP/electrode (P_{Cu} and P_{CNT}) (Figs. R12a and b). It is observed that the magnitude of the Raman peak at $\sim 742\text{ cm}^{-1}$ corresponding to TFSI⁻ ions at P_{CNT} point varied slightly in the first 30s (Figs. R12c

and e). Interestingly, after the 30s, it's clear to find that the amplitude of the peak intensity of TFSI⁻ ions significantly decreased until the 160s, which proves that the concentration of TFSI⁻ ions at P_{CNT} point dropped. As the TFSI⁻ ions near the hot side interface were motivated by heat, these TFSI⁻ ions started departing from the hot side to transport to the cold side. Accordingly, the TFSI⁻ ions diffused away from the hot side toward the cold side and led to a decrease in the concentration of TFSI⁻ ions near the hot interface after a certain period. The intensity of peak at $\sim 742\text{ cm}^{-1}$ reached a steady state value of approximately 60% of the initial status. In contrast, for the Cu|PNP|Cu system, the variation of the magnitude of the TFSI⁻ peak intensity at P_{Cu} region is very limited (Figs. R12d and e), which is much more weakened than the a-CNT|PNP|a-CNT system. The Raman mapping strongly suggests that the TFSI⁻ ion is more active near the a-CNT electrode than that of the Cu electrode, demonstrating the a-CNT electrode is more favourable for the diffusion of TFSI⁻ ions.

Fig. R12 Line-scanning Raman spectra near the interface of (a) a-CNT|PNP|a-CNT system and (b) Cu|PNP|Cu system in a range of 100-1800 cm^{-1} . The intensity variation of the TFSI⁻ peak of (c) a-CNT|PNP|a-CNT and (d) Cu|PNP|Cu system in a range of

710-780 cm^{-1} . (e) The comparison of the intensity variation of the TFSI⁻ peak at 742 cm^{-1} of a-CNT|PNP|a-CNT, and Cu|PNP|Cu system.

Moreover, the DFT calculations of the interaction energy between the electrodes (CNT and Cu) and the ions (TFSI⁻ and Na⁺) are performed by using the Vienna ab initio Simulation Package (VASP) code (*Comput. Mater. Sci.* 6, 15, 1996; *Phys. Rev. B* 54, 11169, 1996; *Phys. Rev. B* 47, 558, 1993), which the exchange-correlation effects are treated by the generalized gradient approximation (GGA) in the Perdew-Burke-Ernzerhof (PBE) parametrization (*Phys. Rev. Lett.* 77, 3865, 1996). A vacuum layer > 30 Å is set on the electrode surfaces. The TFSI⁻ and Na⁺ ions are separately put on the electrode surface and the structures are optimized. To embody the electronic states of ions, the total valence electronic numbers are set to be $N_{\text{all}-1}$ for the models with Na⁺ ion and $N_{\text{all}+1}$ with TFSI⁻ ion, where N_{all} indicates the total intrinsic valence electronic numbers as shown in Fig. R13a. In the geometry optimization, cutoff energy of 500 eV, energy convergence of 10^{-5} eV, and force convergence of 10^{-4} eV/Å are set. The total energy is recorded as $E_{\text{tot,ads}}$. Then, ions moved far away from the electrode by 15 Å, which is enough to exclude the interactions between electrodes and ions, and the total energy is recorded as $E_{\text{tot, far}}$. Lastly, adsorption energy, E_{ad} , can be calculated by $E_{\text{ad}} = E_{\text{tot, far}} - E_{\text{tot, ads}}$. A higher E_{ad} means a stronger interaction between electrodes and ions. From the DFT calculation results (Figs. R13b-e), it's interesting to find that the TFSI⁻ ions formed a stronger interaction with the Cu electrode (0.53 eV) than that of the CNT electrode (0.31 eV). In contrast, the Na⁺ ions formed a stronger interaction with the CNT electrode (1.82 eV) than that of the Cu (1.30 eV) electrode, as summarized in Fig. R13e, which is consistent with previous MD calculation results.

In summary, the *in-situ* Raman measurement and DFT results strongly proved that the TFSI⁻ ions dominated the thermodiffusion process in the a-CNT|PNP|a-CNT system, and the stronger interaction between Na⁺ ions and a-CNT electrode inhibited the diffusion of the Na⁺ ions. As a result, the a-CNT|PNP|a-CNT exhibits the *n*-type characteristic.

We have revised the above discussion in the revised manuscript (Page #9, Line183, Figs. 2a-2f) and Supplementary Information (Page #S11, Figs. S8, S9 and S13) marked with red color.

Fig. R13 The DFT calculation results. (a) The interface models between electrodes and ions in both the Cu|PNP|Cu, and CNT|PNP|CNT systems. The interfacial interaction energy between (b) Cu-TFSI⁻ ions (top) and CNT-TFSI⁻ ions (bottom), (c) Cu-Na⁺ ions (top) and CNT-Na⁺ ions (bottom). (d) The illustration of the symbols of electrodes and ions. (e) The value of the interaction energy between electrodes and ions.

(3) I feel there is a misunderstanding concerning eq. (1) and the subsequent discussion. In fact, eq. (1) does not give the stationary Seebeck coefficient of iTE materials, but rather the transient ion current after switching on the temperature gradient. Thus eq. (1) would explain the initial slope of the cusps occurring in Fig. 1 d,e after each change of ΔT . Perhaps the authors find it helpful to have a look at ref. 18 or at a paper by Janssen and Bier (*Phys. Rev. E* 99, 2019).

Response: We agree that “Eq 1., $S_i = \frac{\sum_i q_i n_i^0 \hat{S}_i D_i}{\sum_i q_i^2 n_i^0 D_i}$ ” in the previously submitted

manuscript describes the response of the transient ion charge to the temperature difference, but it is not suitable for describing the process of ion motion reaching a stationary state. In the paper suggested by the reviewer (*Phys. Rev. E*. 99, 042136, 2019;

J. Chem. Phys. 154, 164511, 2021), and in one of our recent papers (*Cell Rep. Phys. Sci.* 3, 101018, 2022, <https://doi.org/10.1016/j.xcrp.2022.101018>), the ionic Seebeck coefficient of *i*-TE materials is suggested to be related to the timescale. The “Eq. 1” is proper to describe the ionic thermoelectric coefficient when the time reaches the Debye timescale $\tau_D = \lambda^2/D$, which is in a transient state. However, when time evolves to the diffusion timescale, ($\tau_{dif} = 4L^2/(\pi^2D)$, where λ , $2L$ and D are Debye length, the electrode separation, ionic diffusivity, respectively, the τ_{dif} is much longer than the Debye timescale. The “Eq. 1” is not appropriate for explaining the stationary states. For the diffusion timescale, the Seebeck coefficient can be described as *Eq. (R2)*:

$$S_i = \frac{\sum_i q_i e \hat{S}_i n_i^0}{\sum_i (q_i e)^2 n_i^0} \quad (R2)$$

Here, the Seebeck coefficient is strongly related to the Eastman entropy, \hat{S} , (*Cell Rep. Phys. Sci.* 3, 101018, 2022).

Since “*Eq. 1*” only describes the ions in transient status for the *i*-TE materials, it fails to account for the stationary states and doesn’t consider the effect of electrodes. Accordingly, this formula “*Eq. 1*” is not suitable to explain our experiments. We have deleted the discussion related to “*Eq. 1*” and cited the reference suggested by the reviewer in the revised manuscript. As the main finding is that bipolar thermopower of the same materials by using different electrodes, the *in-situ* Raman experimental study together with DFT and MD theoretical analysis results are added to the revised manuscript.

Reviewer #3 (Remarks to the Author):

The author used the interaction between ions and electrodes to control the migration of cations and anions by changing the electrode materials, and achieved positive and negative conversion of the thermopower of PNP films. This reversible p-n transition provides a new approach for the construction of thermodiffusion ionic thermoelectric devices. However, this work is incomplete, and there are many

issues which should be further clarified. I also have some suggestions for the authors to further polish their paper.

(1). The title of this paper says “ionic thermoelectric polymer composite for continuous energy generation”, however, the work on ionic thermoelectric generator is not yet complete. The low power generation of a single PNP film is not meaningful to evaluate its performance. The author needs to connect multiple PNPs in series and consider its stability after many cycles.

Response: As the reviewer suggested, we further investigated the performance of *i*-TEG by connecting more series of PNP films. As shown in Figs. R14a-c, the *i*-TEG consists of 10 pairs of Cu electrodes and CNT electrodes connected in series to improve the output power. Firstly, the 10 pairs of the patterned Cu|Cu electrodes were pressed tightly to contact the 10 pieces of PNP films from above (Fig. R14b). After introducing a temperature difference (ΔT) across 10 pieces of PNP films, taking the data in the first cycle as an example in Fig. 4d, the *i*-TEG produced a negative thermal voltage V_i until reached a relatively stable value after heating (Fig. R14d). Next, the *i*-TEG was connected to a load to output power to the external circuit and the voltage V_i dropped to near zero, which was caused by the accumulation of electrons and holes in the electrodes to balance V_i . Then removing the Cu|Cu electrodes while keeping heating constant, the 10 pairs of patterned a-CNT|a-CNT electrodes were switched to contact with the 10 pieces of PNP films (Fig. R14d) and the external resistor was disconnected simultaneously. It is clear to observe the *i*-TEG produced an opposite sign of thermal voltage (Fig. R14d). Because using a-CNT electrodes make the PNP behave the *n*-type characteristic and the TFSI⁻ anions dominated the thermodiffusion process. Finally, the

i-TEG was connected to the load again to output power to the external load. Furthermore, after repeatedly alternatively switching the patterned a-CNT and Cu electrodes while keeping the heat source constant, the produced thermal voltage of the fabricated *i*-TEG demonstrated high repeatability after 20 cycles (Fig. R14b). Importantly, the proposed *i*-TEG achieves generating cyclic power under a constant heat source without the need to turn on/off the heat source or join/separate materials from the heat source. As a result, there is no necessity to change the thermal contact between the material and heat source and therefore the temperature difference does not need to be re-established, providing a significant innovative impact for expanding the practical applications. Since the cycling stability testing experiments are limited by manual operation, we believe that the cycling stability could be further improved by automatically controlling the processes of exchanging electrodes, which will be investigated in our future work.

We have added the discussion on the latest performance of *i*-TEG with the 10 series of PNP in the revised manuscript (Page #16, Line 347, Figs. 4a and c) and Supplementary Information (Page #S21, Fig. S18).

Fig. R14. (a) The digital photo of the fabricated *i*-TEG by 10 pieces of PNP films alternatively contacting with 10 pairs of Cu|Cu, and a-CNT|a-CNT electrodes. The fabricated *i*-TEG alternatively contacts with (b) Cu|Cu electrodes and (c) a-CNT|a-CNT electrodes. (d) The cyclic working performance of the *i*-TEG under a constant ΔT .

(2). According to Fig. 4, it needs switching the electrodes while also disconnecting the external load and waiting for a certain period to generate the open circuit voltage. I think the generator does not produce power continuously, which is not consistent with the “continuous power generation” claimed in the title of this paper. In addition, there is an error in the labeling sequence of Fig. 4b.

Response: As suggested by the reviewers, we have changed the title “continuous power generation” to “cyclic power generation” in the revised manuscript and Supplementary Information.

Currently, almost all the *i*-TE generators (*i*-TEGs) based on the Soret mechanism operated in a capacitive charge/discharge mode, in which the ions cannot directly transfer across the electrodes but accumulate at the electrode interface under a temperature gradient. Accordingly, the *i*-TEGs cannot produce continuous output

current flow like electron/hole conducting TE materials. It's necessary to periodically switch the on-off state of the heat source or make *i*-TE material attach/detach to the heat source to realize the reciprocating motion of ions (*Nat. Mater.* 18, 608-613, 2019; *Energy Environ. Sci.* 9, 1450-1457, 2016; *Nat. Commun.* 10, 1093, 2019). It remains challenging for *i*-TEGs to generate cyclic power output under a constant heat source.

In this work, we proposed a new operation principle for cyclic generating power based on the discovery of bipolar thermoelectricity of the same *i*-TE material. The proposed *i*-TEG always contacts the constant heat source at bottom, as shown in Fig. R15. It realizes cyclic power generation by alternately contacting different electrodes from above (Cu|Cu, and a-CNT|a-CNT). Meanwhile, after further optimization, we have shortened the waiting time and sped up the rate of switching the Cu|Cu and a-CNT|a-CNT electrodes can help reduce the interruption time. Yes, it is unavoidable to have the process of exchanging electrodes, which indeed brings interruption in the power generation process for a certain period. We think that further improving the conductivity of the *i*-TE materials leads to fast transportation of ions, which could significantly shorten the period for generating thermal voltage.

Meanwhile, according to the comment “*In addition, there is an error in the labeling sequence of Fig. 4b.*”, we have corrected the typo of the wrong sequence label of “Fig. 4b”, as shown in Fig. R16. The revision parts are labeled in red color in the revised manuscript (Page #18, Fig. 4b).

Fig. R15 The diagram illustration of the working mechanism of the *i*-TEG with a constant ΔT by alternately exchanging the Cu|Cu, and a-CNT|a-CNT electrodes. The schematic illustration of the top view of the PNP *i*-TE material contacting with (a) Cu|Cu electrodes and (b) a-CNT|a-CNT electrodes. (c) The schematic illustration of working principle of the designed *i*-TEG when alternately switching the electrodes.

Fig. R16 The illustration of the working procedures of the *i*-TEG under a constant ΔT by alternately exchanging the Cu|Cu electrodes with the a-CNT|a-CNT electrodes and the corresponding diagrams of the transportation process of the cations and anions.

(3). For the measurement of the Seebeck coefficient, the connection between the positive and negative poles of the voltmeter and the hot and cold ends of materials will affect the positive and negative values of voltages. Therefore, please supply the measurement details, schematic and physical diagram of measurement equipment for the Seebeck coefficient. Moreover, in Fig. 1d, the relationship between the thermoelectric voltage and temperature difference of p-type Cu|PNP|Cu does not correspond to Fig. S2. The same problem exists in Fig. 1e.

Response: Yes, as the reviewer suggested, the detailed diagram of measurement equipment of the ionic thermopower measurement setup is in Fig. R17 and the detail of the measurement procedure is as followed. The tested *i*-TE materials with a rectangular shape were suspended on two electrodes with a separation distance of several tens of millimeters. And one Peltier heater and one Peltier cooler were located under the tested materials, powered by two Keithley 2400 source meters, providing temperature difference along the length direction of PNP. The produced thermal voltages by the *i*-TE materials were recorded with a nano voltage meter (Keithley 2182A). Two T-type thermocouples were applied to record temperature variation between the hot and cold sides of *i*-TE materials controlled by national instruments (NI) mode 9213 coupled with NI 9162 mode. The whole test setup of the measurement system was first calibrated well with the reported *i*-TE materials in the previous work before whole measurement (*Nat. Commun.* **10**, 1093, 2019; *Adv. Energy Mater.* **9**, 1901085, 2019).

Fig. R17. The schematic of the homemade ionic Seebeck coefficient measurement setup. The red and blue regions denote the heat source and heat sink, respectively.

Moreover, according to the reviewer's comment "*Moreover, in Fig. 1d, the relationship between the thermoelectric voltage and temperature difference of p-type Cu|PNP|Cu does not correspond to Fig. S2. The same problem exists in Fig. 1e*", we have carefully checked the thermoelectric voltage and temperature difference of p-type Cu|PNP|Cu in Fig. 1d and Fig. S2 as well as n-type a-CNT|PNP|a-CNT in Fig. 1e and Fig. S2 (Supplementary Information of the previously submitted manuscript). **We found the coordinate in the Y-axis in Fig. S2 (Supplementary Information of the previously submitted manuscript) should be " $-\Delta V (V_{\text{hot}}-V_{\text{cold}})$ ", which missed the negative symbol "-",** as shown in Figs. R18 c-d. According to the definition of the ionic Seebeck coefficient (S_i) in Eq. R1

$$S_i = -\frac{V(T_H) - V(T_C)}{T_H - T_C} \quad (\text{R1})$$

where $V(T_H)$ and $V(T_C)$ correspond to the voltage of the hot electrode at temperature T_H and the cold electrode at temperature T_C , respectively, (*Sci. Adv.* **7**, eabi7233 2021; *J. Chem. Phys.* **154**, 164511, 2021).

When started heating, the hot-electrode and cold-electrode sides were connected to the positive and negative electrodes of the voltage meter (Keithley 2182 nano-voltage meter), respectively. The produced voltage difference between the hot side and cold side of Cu|PNP|Cu ΔV_{Cu} ($V_{hot}-V_{cold}$) gave a negative value (Fig. R18a), indicating a higher amount of positive Na^+ cations moved to the cold side, performing *p*-type characteristics. Then, the Cu|Cu electrodes were replaced by the a-CNT|a-CNT electrodes, and the connection between a-CNT|PNP|a-CNT systems with voltage meter is exactly the same as the Cu|PNP|Cu systems. The measured ΔV_{CNT} produced a positive sign (Fig. R18b), implying TFSI⁻ anions dominated the thermodiffusion, belonging to the *n*-type characteristic. By fitting the slope of the “ $-\Delta V_{Cu}$ vs ΔT ” or “ $-\Delta V_{CNT}$ vs ΔT ”, the measured thermopower of Cu|PNP|Cu and a-CNT|PNP|a-CNT are 20 mV K⁻¹ and -10.64 mV K⁻¹, respectively. We found the coordinate in the *Y*-axis should be “ $-\Delta V$ ” instead of “ ΔV ”, which is revised accordingly as shown in Figs. R18 c-d.

We have added the detail of the measurement setup and corrected the typo marked with red color in the manuscript and the Supplementary Information (Pages #S4 and #S26, Fig. S1).

Fig. R18 The measured produced thermal voltage of (a) Cu|PNP|Cu and (b) a-CNT|PNP|a-CNT systems under various temperature differences. The fitting curves of $-\Delta V$ vs ΔT of (c) Cu|PNP|Cu and (d) a-CNT|PNP|a-CNT systems.

(4) *What are the advantages of VA-CNT as electrodes compared to other carbon materials such as SWCNT or MWCNT? The reason for choosing VA-CNT arrays as electrodes should be described.*

Response: As suggested by the reviewer, the physical properties of aligned CNT were further described. To avoid misunderstanding, we have changed the name “vertical aligned CNT” to “aligned CNT (a-CNT)” in the revised manuscripts. As the aligned CNT films were placed horizontally, and the length direction of the aligned CNT was parallel to the *i*-TE polymer material when performing test, which is not vertical to the polymer material.

Advantages: Firstly, the aligned CNT thin film is commercially available with good quality control and consistency, excellent property repeatability and stability (Nanjing Ji Cang Nano Technology Co., Ltd. (Nanjing, China), and the product number is JCNTF-20C). Meanwhile, the aligned CNTs consist of bundles of carbon tubes arranged orderly along the length direction with small entanglements, they exhibited higher conductivity of $3 \times 10^5 \text{ S m}^{-1}$, a lower sheet resistance of $1 \sim 1.5 \text{ } \Omega \text{ m}^{-2}$, excellent flexibility, low cost, and stronger mechanical strength of $\sim 120 \text{ MPa}$ compared to common SWCNT and MWCNT-based films, as summarized in Table R1. Moreover, due to the softness and self-adhesive nature of the aligned CNT, it is easy to form tight contact with the PNP composite compared to thick SWCNTs and MWCNTs films. Having good contact is of great significance for reducing contact resistance and improving the overall ionic thermoelectric conversion performance. Compared to the thermoelectric property of PNP tested with the MWCNT and SWCNT electrodes, the a-CNT|PNP|a-CNT performed higher thermopower of $-10.2 \pm 0.83 \text{ mV K}^{-1}$ than SWCNT|PNP|SWCNT ($-5.2 \pm 1.35 \text{ mV K}^{-1}$) and MWCNTs ($-8.17 \pm 1.2 \text{ mV K}^{-1}$).

We have added the discussion of the advantages of aligned CNTs as the electrodes marked with red color in the revised manuscript (Page #7, Line 139) and Supplementary Information (Page #S8, Fig. S5, Table S1).

Table R1 The comparison of the physical property of the three CNTs.

Property	Thickness	Conductivity	Sheet resistance	Strength	Length	Diameter
aligned CNT	~10 μm	$\sim 3 \times 10^5 \text{ S m}^{-1}$	1~1.5 $\Omega \text{ m}^{-2}$	~120 MPa	continuous growth	20-50 nm
MWCNT	~0.15 mm $\pm 0.05 \text{ mm}$	/	~4 $\Omega \text{ m}^{-2}$	~10 MPa	10-30 μm	20-30 nm
SWCNT	60-80 μm	$2 \sim 4 \times 10^3 \text{ S m}^{-1}$	/	10-15 MPa	10-30 μm	1-2 nm

(5). *There is an error in the figure caption of Fig. 2a and Fig. 2b, which is inconsistent with the description in line 185 on page 10.*

Response: The Figure caption of Fig. 2a and Fig. 2b in the submitted manuscript should be “the top view MD snapshots of NaTFSI molecules adsorbed on (a) Cu and (b) CNT electrodes surface at 300K”, as shown in Fig. R19.

We have corrected the error in the caption of “Figs. 2a and 2b” to make it consistent with the description marked with red color in the revised manuscript and Supplementary Information (Page #14, Fig. S11).

Fig. R19 Top view MD snapshots of NaTFSI molecules adsorbed on (a) Cu and (b) aligned CNT electrode surface at 300K.

(6). *The ionic thermoelectric polymer was described as all-solid-state i-TE material. However, this i-TE material contains liquid propylene carbonate (PC), so it is not*

considered to be an all-solid material.

Response: In this work, the reported PNP consists of polymer matrix PVDF-HFP, salt NaTFSI, and the low-molecular-weight PC. The PC acted as a plasticizer to enlarge the amorphous part of PVDF-HFP to obtain high ion mobility. At the same time, the obtained PNP film still maintains solid-state features as shown in Fig. R20. It is different from the ion gels which contained a large amount of ionic liquid solutions.

As the reviewer suggested, we have revised the term “all-solid-state” to “solid-state” marked with red color in the revised manuscript to avoid misleading.

Fig. R20. The digital photo of the prepared solid-state PNP film.

REVIEWERS' COMMENTS

Reviewer #1 (Remarks to the Author):

My concerns were well addressed in the revised manuscript. I recommend accepting the manuscript.

Reviewer #2 (Remarks to the Author):

The authors have addressed all points, - their reply is overall satisfactory.

Reviewer #3 (Remarks to the Author):

My concerns have been properly addressed. The manuscript can be accepted as it is.

Revision Report

Response to the comments by the referees on the manuscript (*NCOMMS-22-27100A*) entitled “by C. Chi[†], G. Liu[†], M. An[†], Y. Zhang, D. Song, X. Qi, C. Zhao, Z. Wang, Y. Du, Z. Lin, Y. Lu, H. Huang, Y. Li, C. Lin, W. Ma*, B. Huang*, X. Du, and X. Zhang”.

We appreciate the editor and the referees for their careful consideration, insightful comments, constructive suggestions, and positive evaluation of improving our manuscript. Accordingly, we have carefully revised our manuscript according to the referees’ suggestions. All the revised and added parts have been marked in red color in the manuscript. Below are the point-by-point responses to the referee’s comments.

Comments from the editors and referees:

Reviewer #1 (Remarks to the Author):

My concerns were well addressed in the revised manuscript. I recommend accepting the manuscript.

Response: We thank the reviewer again for the careful examination and the positive evaluation of our manuscript.

Reviewer #2 (Remarks to the Author):

The authors have addressed all points, their reply is overall satisfactory.

Response: We thank the reviewer again for the careful examination and the positive evaluation of our manuscript.

Reviewer #3 (Remarks to the Author):

My concerns have been properly addressed. The manuscript can be accepted as it is.

Response: We thank the reviewer again for the careful examination and the positive evaluation of our manuscript.

We have edited the format of the manuscript and Supplementary Information according to the editorial suggestions.

Many thanks to you for your time and knowledge.